# Environment determines evolutionary trajectory in a constrained phenotypic space

David T Fraebel[1,2], Harry Mickalide[1,2], Diane Schnitkey[1,2], Jason Merritt[1,2], Thomas E Kuhlman[1,2,3], Seppe Kuehn[1,2,3]*

[1]Center for the Physics of Living Cells, University of Illinois at Urbana-Champaign, Urbana, United States; [2]Department of Physics, University of Illinois at Urbana-Champaign, Urbana, United States; [3]Center for Biophysics and Quantitative Biology, University of Illinois at Urbana-Champaign, Urbana, United States

**Abstract** Constraints on phenotypic variation limit the capacity of organisms to adapt to the multiple selection pressures encountered in natural environments. To better understand evolutionary dynamics in this context, we select *Escherichia coli* for faster migration through a porous environment, a process which depends on both motility and growth. We find that a trade-off between swimming speed and growth rate constrains the evolution of faster migration. Evolving faster migration in rich medium results in slow growth and fast swimming, while evolution in minimal medium results in fast growth and slow swimming. In each condition parallel genomic evolution drives adaptation through different mutations. We show that the trade-off is mediated by antagonistic pleiotropy through mutations that affect negative regulation. A model of the evolutionary process shows that the genetic capacity of an organism to vary traits can qualitatively depend on its environment, which in turn alters its evolutionary trajectory.

*For correspondence: seppe@ illinois.edu

**Competing interests:** The authors declare that no competing interests exist.

## Introduction

In nature organisms adapt to complex environments where many biotic and abiotic factors affect survival. For microbes these factors include demands on metabolism (*Savageau, 1983*), motility (*Celani and Vergassola, 2010*) and antibiotic resistance (*Vetsigian et al., 2011*). In this context, evolution involves the simultaneous adaptation of many phenotypic traits. Organisms under complex selection pressures often cannot vary traits independently and instead exhibit trade-offs (*Shoval et al., 2012*).

Trade-offs constrain adaptive responses to selection. For example, phage exhibit a trade-off between fecundity and virulence which depends on the relative duration of periods of horizontal and vertical transmission (*Messenger et al., 1999*). Bacterial populations selected for efficient conversion of nutrients to biomass exhibit a trade-off between yield and growth rate (*Bachmann et al., 2013*).

Predicting evolution in complex environments requires quantifying both trade-offs and selection pressures (*Lande, 1979*). In wild populations of birds (*Grant and Grant, 1995*) and fish (*Ghalambor et al., 2003*), phenotypic constraints and selection pressures have been inferred from measurements of phenotypic variation. However, in wild populations of higher organisms it is challenging to observe evolution, determine selection pressures and elucidate mechanisms constraining phenotypes. To better understand the interplay between trade-offs, selection and evolution, it is necessary to study genetically tractable, rapidly evolving microbial populations in the laboratory.

However, laboratory-based experimental evolution of microbes typically selects for a single phenotype such as growth rate (*Lang et al., 2013*). There is evidence that metabolic trade-offs arise in

**eLife digest** In nature organisms face many challenges, and species adapt to their environment by changing heritable traits over the course of many generations. How organisms adapt is often limited by trade-offs, in which improving one trait can only come at the expense of another.

In the laboratory, scientists use well-controlled environments to study how populations adapt to specific challenges without interference from their natural habitat. Most experiments, however, only look at simple challenges and do not take into account that organisms in the wild face many pressures at the same time. Fraebel et al. wanted to know what happens when an organism's performance depends on two traits that are restricted by a trade-off. The experiments used populations of the bacterium *Escherichia coli,* which can go through hundreds of generations in a week, providing ample opportunity to study mutations and their impact on heritable traits.

Through a combination of mathematical modeling and experiments, Fraebel et al. found that the environment is crucial for determining how bacteria adapt when their swimming speed and population growth rate are restricted by a trade-off. When nutrients are plentiful, *E. coli* populations evolve to spread faster by swimming more quickly despite growing more slowly. Yet, if nutrients are scarcer, the bacteria evolve to spread faster by growing more quickly despite swimming more slowly. In each scenario, the experiments identified single mutations that changed both swimming speed and growth rate by modifying regulatory activity in the cell.

A better understanding of how an organism's genetic architecture, its environment and trade-offs are connected may help identify the traits that are most easily changed by mutations. The ultimate goal would be to be able to predict evolutionary responses to complex selection pressures.

these experiments from the decay of traits that are not subject to selection (*Cooper and Lenski, 2000*) rather than a compromise between multiple selection pressures. Other experiments explore how phenotypes restricted by trade-offs evolve under alternating selection for individual traits (*Yi and Dean, 2016*; *Messenger et al., 1999*). Less is known about evolutionary dynamics in the naturally relevant regime where selection pressures are multifaceted.

To address this, we selected *Escherichia coli* for faster migration through a porous environment. We showed that the evolution of faster migration is constrained by a trade-off between swimming speed and growth rate. Evolution of faster migration in rich medium is driven by faster swimming despite slower growth, while faster migration in minimal medium is achieved through faster growth despite slower swimming. Sequencing and genetic engineering showed that this trade-off is due to antagonistic pleiotropy through mutations that affect negative regulation. Finally, a model of multi-trait selection supports the hypothesis that the direction of evolution when phenotypes are constrained by a trade-off is determined by the genetic variance of each trait. Our results show that when selection acts simultaneously on two traits governed by a trade-off, the environment determines the evolutionary trajectory.

## Results

### Experimental evolution of migration rate

*E. coli* inoculated at the center of a low viscosity agar plate consume nutrients locally, creating a spatial nutrient gradient which drives chemotaxis through the porous agar matrix (*Righetti et al., 1981*; *Maaloum et al., 1998*) and subsequent nutrient consumption (*Adler, 1966*; *Wolfe and Berg, 1989*; *Croze et al., 2011*). As a result, the outermost edge of the expanding colony is driven by both growth and motility (*Koster et al., 2012*). The result is a three-dimensional bacterial colony that expands radially across the plate as individuals swim and divide in the porous environment. We refer to the outermost edge of an expanding colony as the migrating front. We tracked these migrating fronts using webcams and light-emitting diode (LED) illumination (Materials and methods). The front migrates at a constant speed $s$ after an initial growth phase (*Adler, 1966*; *Wolfe and Berg, 1989*).

We performed experimental evolution by repeating rounds of allowing a colony to expand for a fixed time interval, selecting a small population of cells from the migrating front and using them to

inoculate a fresh low viscosity agar plate (*Figure 1a*). By isolating cells from the migrating front, our procedure selects both for motility and growth rate. We performed selection experiments in this way for two distinct nutrient conditions. First, we used rich medium (lysogeny broth (LB), 0.3 % w/v agar, 30°C) where all amino acids are available. In this medium the population forms concentric rings (*Figure 1b*) that consume amino acids sequentially. The outermost ring consumes L-serine and most of the oxygen (*Adler, 1966*). Second, we used minimal medium (M63, 0.18 mM galactose, 0.3 % w/v agar, 30°C) where populations migrate towards and metabolize galactose with a single migrating front.

In rich medium, colonies of wild-type bacteria (MG1655-motile, founding strain) expand with a front migration speed $s \approx 0.3$ cm h$^{-1}$ and cells were sampled from the front after 12 hr (*Figure 1b*). A portion of this sample was used to immediately inoculate a fresh plate while the remainder was preserved cryogenically. The process was repeated every 12 hr for 15 rounds. We observed a nearly 50% increase in $s$ over the course of the first 5 rounds of selection. The increase in $s$ was largely reproducible across five independent selection experiments (*Figure 1c*). We estimate that plate-to-plate variation in agar concentration due to evaporative loss could change the migration rate by up to 0.06 cm h$^{-1}$ in later rounds (Appendix 1). However, independent replicate selection experiments exhibit fluctuations in migration rate that exceed this estimate. For example, replicate 4 declines in later rounds of selection, and this decline may reflect the unique low abundance mutation that appears in this replicate by round 15 (Figure 5a). In addition, replicate 3 exhibits substantially faster migration than replicates 1, 2 and 4 in round 7, and this may reflect the distinct mutations observed in this replicate at round 5 (Figure 5a). So, while migration rates increased in all replicates, the magnitude of the increase differed between replicates.

To check whether chemotaxis was necessary for increasing $s$, we performed selection experiments using a motile but non-chemotactic mutant ($\Delta cheA$-$Z$, Materials and methods). Motility in this strain was confirmed by single-cell imaging in liquid media. As observed previously (*Wolfe and Berg, 1989*), the non-chemotactic strain formed dense colonies in low viscosity agar that remained localized near the site of inoculation and expanded ~1 cm in a 24 hr period: a rate 10-fold slower than the wild-type. To allow sufficient time for colony expansion, we performed selection experiments using this strain with 24 hr incubation times and observed an increase in $s$ from approximately 0.03 cm h$^{-1}$ to 0.04 cm h$^{-1}$ (*Figure 1—figure supplement 1*). We did not observe fast migrating spontaneous mutants which have been reported previously in multiple species (*Wolfe and Berg, 1989*; *Mohari et al., 2015*), likely because our plates were incubated for a shorter period of time.

To determine the number of generations transpiring in our selection experiments, we measured the number of cells in the inoculum and the number of cells in the colony after 12 hr of growth and expansion (Materials and methods). We estimated that 10 to 12 generations occurred in each round of selection. We then tested whether prolonged growth in well mixed liquid medium for a similar number of generations could lead to faster migration by growing the founding strain for 200 generations in continuous liquid culture and periodically inoculating a low viscosity agar plate (*Figure 1—figure supplement 2*). We observed only a 3.5% increase in the rate of migration, demonstrating that selection performed on spatially structured populations results in more rapid adaptation for fast migration than growth in well mixed conditions.

We then performed selection experiments in a minimal medium where growth and migration are substantially slower than in rich medium (*Figure 1d*). In this condition we allowed 48 hr for each round of expansion and took precautions to limit evaporative loss in the plates over this longer timescale (Materials and methods). In the first round, the population formed small ~1.5 cm diameter colonies without a well defined front. Populations formed well defined fronts in subsequent rounds of selection (*Figure 1d*), reflecting a transition from growth and diffusion dominated transport to chemotaxis dominated migration (*Croze et al., 2011*). We observed an approximately 3-fold increase in $s$ over the course of 10 rounds of selection. Variation across replicate experiments was substantial, and exceeded our estimate of systematic error due evaporative losses changing the agar concentration (Appendix 1). So while all replicates increased their migration rate, the magnitude of the increase in migration rate varied substantially. This variation may be due to the different mutations present across replicates (Figure 5b).

When we performed selection in minimal medium using the non-chemotactic mutant ($\Delta cheA$-$Z$), we found little or no migration and only a very small increase in the migration rate over 10 rounds of

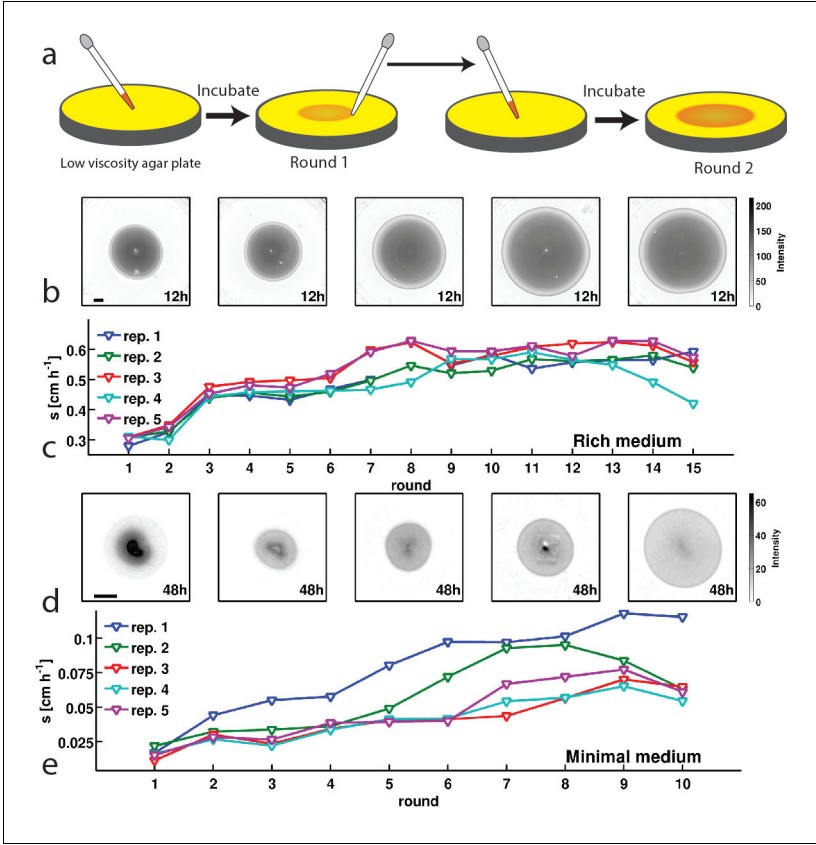

**Figure 1.** *E. coli* evolves faster migration through a porous environment in rich and minimal media. (a) A schematic of the selection procedure. *E. coli* are inoculated into the center of a low viscosity (0.3 % w/v) agar plate where they form an expanding colony driven by metabolism and motility. After a fixed period of incubation, samples are taken from eight locations around the outer edge of the expanded colony, mixed, and used to inoculate a fresh plate. (b) Shows expanded colonies in rich medium (LB) plates after 12 hr of incubation over five successive rounds of selection. The color bar to the right applies to all panels in (b), with darker gray indicating higher cell density. Image intensity is assumed to be monotonic but not linear with cell density in the plate. Scale bar in the left panel is 1 cm and applies to all panels in (b). (c) Shows the rate of migration as a function of round of selection over 15 rounds for five replicate selection experiments in rich medium. No rate is reported for replicate 1 round 8 due to failure of the imaging device. Errors in measured rates of migration are smaller than the size of the markers. (d) Shows colonies (gray regions) in minimal medium (M63, 0.18 mM galactose) after 48 hr of incubation. The color bar to the right applies to all panels in (d). The scale bar in the left panel is 1 cm. (e) Shows the rate of migration as a function of round of selection over 10 rounds for five replicate selection experiments in minimal medium. Errors in migration rates were smaller than the size of markers. See Materials and methods for details of image processing in both experiments.

The following figure supplements are available for figure 1:

**Figure supplement 1.** Selection with non-chemotactic (ΔcheA-Z) mutant.

**Figure supplement 2.** Change in migration rate during long-term liquid culture.

**Figure supplement 3.** Adaptation in rich medium depends on sampling location.

**Figure supplement 4.** Comparison of founding and evolved strains to RP437.

**Figure supplement 5.** Persistence of rich medium fast migrating phenotype in liquid culture.

selection (*Figure 1—figure supplement 1*). We concluded that chemotaxis is also necessary for increasing $s$ in this medium.

Using the same technique described for rich medium, we estimated the number of generations per round of selection in minimal medium to be <10. We tested whether approximately 120 generations of growth in liquid was sufficient to evolve faster migration in minimal medium. Here we found that prolonged growth in well mixed conditions resulted in ~2-fold faster front migration. Despite the increase in migration rate, selection in well mixed conditions resulted in slower migration than selection in low viscosity agar plates for a similar number of generations (*Figure 1—figure supplement 2*).

## Increasing swimming speed and growth rate increase migration rate

To characterize the adaptation we observed in *Figure 1c,e*, we studied a reaction-diffusion model of migrating bacterial fronts of the type pioneered by *Keller and Segel (1971)* and reviewed in *Tindall et al. (2008)*. We model the bacterial density $\rho(\mathbf{r}, t)$ and a single chemo-attractant that also permits growth $c(\mathbf{r}, t)$. Our model includes only a single nutrient since the growth and chemotaxis of the outermost ring in rich media is driven by L-serine (*Adler, 1966*) and our minimal media conditions contain only a single carbon source/attractant. The dynamics of $\rho(\mathbf{r}, t)$ and $c(\mathbf{r}, t)$ are governed by

$$\frac{\partial \rho}{\partial t} = D_b \nabla^2 \rho - \nabla \cdot \left( \frac{k_0 K_D}{(K_D + c)^2} \rho \nabla c \right) + g(\rho, c) \tag{1}$$

and

$$\frac{\partial c}{\partial t} = D_c \nabla^2 c - f(\rho, c), \tag{2}$$

where the spatial and temporal dependence of $\rho$ and $c$ have been suppressed for clarity. The three terms on the right hand side of *Equation 1* describe diffusion, chemotaxis and growth respectively. $D_b$ is the bacterial diffusion constant, which describes the rate of diffusion of bacteria due to random, undirected motility. $k_0$ is the chemotactic coefficient, which captures the strength of chemotaxis in response to gradients in attractant. $K_D$ is the equilibrium binding constant between the attractant and its associated receptor (*Brown and Berg, 1974*). Growth is modeled using the Monod equation $g(\rho, c) = \frac{k_g \rho c}{K_g + c}$, where $k_g$ is the maximum growth rate and is the concentration of nutrient allowing half-maximal growth. $f(\rho, c)$ describes the nutrient consumption and has an identical form to $g(\rho, c)$ since we assume the yield ($Y$, cells mL$^{-1}$mM$^{-1}$) is a constant. $D_c$ is the diffusion constant of small molecules in water. The physiological parameters describing growth and attractant-receptor binding ($k_g$, $K_g$, $Y$ and $K_D$) were either measured here or have been reported in the literature and can be applied directly in our simulation of migration in both nutrient conditions. *Table 1* describes each parameter used in this study.

The bacterial diffusion constant and the chemotactic coefficient depend on motility and the physical structure of the agar matrix. Motility in *E. coli* consists of runs, segments of nearly straight swimming ~0.5 to 1 s long at ~20 μm s$^{-1}$, and tumbles that rapidly reorient the cell over a period of ~0.1 s (*Berg and Brown, 1972*). Rivero *et al.* showed how the reaction-diffusion parameters $D_b$ and $k_0$ depend on run speed and duration (*Rivero et al., 1989*). *Croze et al. (2011)* modified these results to account for the presence of the agar matrix. The approach treats interactions between cells and agar as scattering events where the cell is forced to tumble.

We estimated $D_b$ and $k_0$ using the method developed by Croze *et al.* for our conditions. With these parameters we simulated the model in *Equations 1 and 2* with parameters appropriate for rich media (chemotaxis towards L-serine) and minimal media (chemotaxis towards galactose). For the founder strain, these simulations predicted a migration rate of 0.61 cm h$^{-1}$ for rich media and 0.08 cm h$^{-1}$ for minimal media compared to measured rates of 0.30 ± 0.01 cm h$^{-1}$ and 0.0163 ± 0.0038 cm h$^{-1}$ respectively. We note that this comparison involves no free parameters.

In rich medium our model describes the dynamics of a single metabolite/attractant (L-serine), and therefore fails to account for secondary fronts behind the outermost front, which arise from the metabolism of other amino acids (*Adler, 1966*) (*Figure 1b*, *Figure 2—figure supplement 2*). This is a reasonable approximation since we select cells only from the outermost front of the colony. In

minimal medium, where only a single nutrient is available, we observe only a single migrating front as our model predicts (*Figure 2—figure supplement 2*). Other limitations of this model include the fact that it does not describe the process of adaptation by chemoattractant receptors (*Berg and Tedesco, 1975*), nor does it describe stochastic processes at the single-cell level such as trapping in the agar matrix and cell-to-cell variability. The discrepancy between predicted migration rate and our observed migration rate most likely arises from the fact that cells are transiently trapped in the agar matrix (*Wolfe and Berg, 1989*) rather than simply being scattered. While more sophisticated models have been developed to include these processes (*Vladimirov et al., 2008*; *Frankel et al., 2014*), the model in *Equations 1, 2* captures the essential features of bacterial front migration with fewer adjustable parameters. See Appendix 1 for further discussion.

To understand how changes in motility and growth could contribute to the evolution of migration, we studied how the migration rate ($s$) varied with the parameters of our model through numerical simulation (Appendix 1). We found that increases in run speed ($|v_r|$) and growth rate ($k_g$) had the largest impact on $s$ (*Figure 2*). Consistent with previous reports, our model indicates that only small gains in migration rate can be achieved through increases in tumble frequency (*Wolfe and Berg, 1989*) ($\sim 10\%$, *Figure 2—figure supplement 3*).

*Figure 2* shows how the front migration rate (heatmap) varies with run speed and growth rate for both nutrient conditions studied in *Figure 1*. Our model predicts that the fastest migrating strain should be the one that increases both its run speed and growth rate relative to the founder. Therefore, in the absence of any constraints on accessible phenotypes, we expect both run speed and growth rate to increase with selection.

## A trade-off constrains the evolution of faster migration

To test the predictions of the reaction-diffusion model, we experimentally interrogated how the motility and growth phenotypes of our populations evolved over the course of selection. We performed single-cell tracking experiments using a microfluidic method similar to one described previously (*Jordan et al., 2013*). This method permitted us to acquire 5 min swimming trajectories from

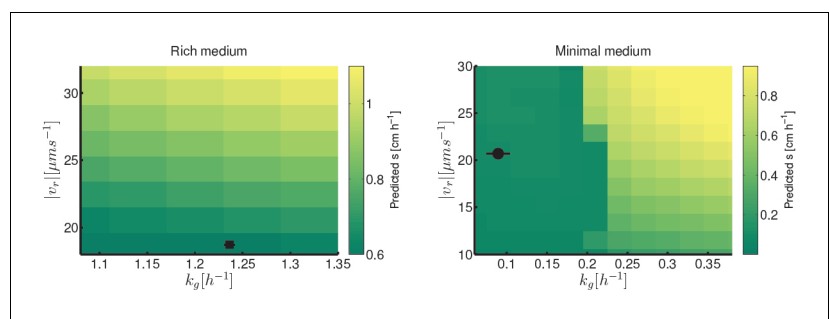

**Figure 2.** Migration rate increases with run speed and growth rate. (a) Front migration rate (heatmap) as a function of run speed ($|v_r|$) and maximum growth rate ($k_g$) simulated using the reaction-diffusion model discussed in the text with parameters appropriate for rich medium conditions (*Table 1*). Model parameters were estimated using the method developed by Croze et al (Appendix 1). Black square shows the run speed and growth rates measured for the founding strain in rich medium (*Figure 3*). Standard error in $|v_r|$ is smaller than the size of the marker; error bar in $k_g$ is the standard deviation across three replicate measurements. (b) Identical to panel (a) except for minimal medium. The abrupt change in migration rate around $k_g = 0.2 \, \text{h}^{-1}$ corresponds to a transition from diffusion dominated front migration to a traveling wave (Appendix 1). The founding strain's phenotype is shown as a black circle, error bars are constructed identically to those in (a).

The following figure supplements are available for figure 2:

**Figure supplement 1.** Reaction-diffusion model recapitulates qualitative features of colony expansion.

**Figure supplement 2.** Comparison of front profiles from simulation and experiment.

**Figure supplement 3.** Simulation of migration rate versus tumble frequency.

hundreds of individuals from strains isolated prior to selection (founder) and after 5, 10 and 15 rounds of selection in rich media (replicate 1, *Figure 1c*) and for the founder and strains isolated after 5 and 10 rounds of selection in minimal media (replicate 1, *Figure 1e*). For tracking, cells were grown in the medium in which they were selected. This technique permitted us to capture more than 280,000 run-tumble events from approximately 1500 individuals. Tracking code is available (*Mickalide et al., 2017*).

We identified run and tumble events for all individuals (*Berg and Brown, 1972*; *Taute et al., 2015*) (Materials and methods). *Figure 3a–b* shows that run durations declined over the course of selection in both rich and minimal media. We show the complementary cumulative distribution function ($c(\tau_r)$) of run durations ($\tau_r$) aggregated across all run events detected for the founding or evolved strains ($c(\tau_r) = 1 - \int_{-\infty}^{\tau_r} d\tau_r' P(\tau_r')$, where $P(\tau_r')$ is the distribution of run durations). $c(\tau_r)$ quantifies the fraction of all runs longer than a time $\tau_r$. These distributions show that the evolved strains exhibited a reduction in the probability of executing long runs. We observed opposite trends for tumble duration, with decreasing tumble duration in rich medium and increasing duration in minimal medium (*Figure 3—figure supplement 2*). To summarize these changes in run-tumble statistics, we computed the tumble bias (fraction of time spent tumbling) and the tumble frequency (tumbles per second, *Figure 3c–d*). In both conditions, we observe an increase in the tumble frequency. This is expected since previous studies showed that mutants with increased tumble frequencies have faster migration rates through agar, likely due to tumbles freeing cells from being trapped in the agar (*Wolfe and Berg, 1989*). In rich medium we observed a decline in tumble bias, while selection in minimal medium increased the tumble bias. Tumble bias and frequency are reported in *Table 2* for all tracked strains.

*Figure 3e–f* show the probability distributions of run speeds for founding and selected strains in both nutrient conditions. In rich medium we observed a nearly 50% increase in the run speed ($|v_r|$) between founder and rounds 10 to 15. Tracking strains isolated after 15 rounds from independent selection experiments (replicates 3 and 4, *Figure 1c*) showed that this increase in run speed was reproducible across independent evolution experiments (*Figure 3—figure supplement 3*). Finally, to check that the phenotype we observed after 15 rounds of selection in rich medium was distinct from standard laboratory strains used in chemotaxis studies, we tracked RP437 and found that its swimming speed was slower than the round 15 strain (*Figure 1—figure supplement 4*).

Surprisingly, when we performed single-cell tracking for strains evolved in minimal media we observed the opposite trend. In these conditions we observed a 50% reduction in run speed (*Figure 3f*). Again, we found that this result was reproducible across independently evolved strains (*Figure 3—figure supplement 3*).

While the overall trend in minimal medium was towards reduced run duration, one replicate showed an increase in run duration (*Figure 3—figure supplement 3*). The strain where we observed long runs after 10 rounds of selection (replicate 2, *Figure 1e*) also exhibited a slower migration rate than the strain isolated from replicate 1, and the long run durations may be responsible for this difference.

We then measured the growth rates of founding and evolved strains from both selection conditions in well mixed liquid corresponding to the medium used for selection (Appendix 1). We observed a decline of about 10% in the maximum growth rate with selection in rich medium and a three-fold increase in the maximum growth rate after 10 rounds of selection in minimal medium (*Figure 3g-h*). We found that these changes in growth rate are reproducible across independently evolved strains in both environmental conditions (*Figure 3—figure supplement 3*).

Since motility is known to depend on the growth history of the population (*Staropoli and Alon, 2000*), we checked whether the phenotypic differences between founding and evolved strains shown in *Figure 3* remained valid when we tracked cells over a range of optical densities during population growth. We performed these measurements for the founding strain in both rich and minimal media, and for a round 15 strain in rich medium and a round 10 strain in minimal medium (*Figure 3—figure supplement 4*). For both rich and minimal media, we found that the differences in run speed ($|v_r|$) between founding and selected strains were retained across the growth curve (*Figure 3—figure supplement 4d*). Likewise, in minimal medium, the average run duration was shorter for the selected strain than for the founder across the growth curve. For rich medium, average run durations for the round 15 strain were not consistently shorter than founder, but the round 15 strain exhibited smaller variability in run duration (*Figure 3—figure supplement 4b*).

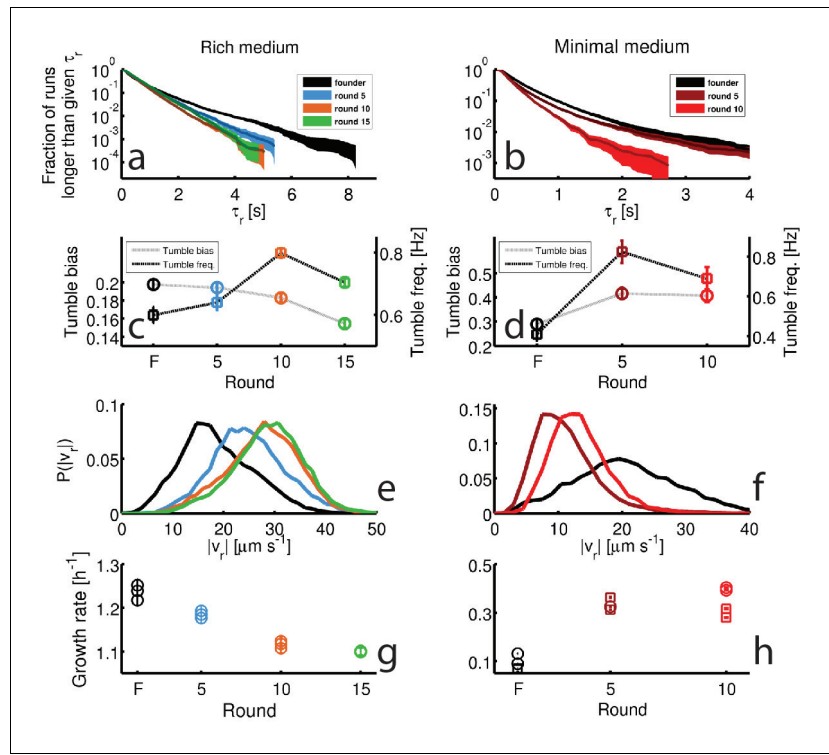

**Figure 3.** Dynamics of phenotypic evolution in rich and minimal media. (**a–f**) Show single-cell swimming phenotypes (run duration ($\tau_r$), run speed ($|v_r|$), tumble bias and tumble frequency, see Materials and methods). Tracking was performed for founding strain (140 cells, 19,597 run events), strains isolated after 5 (79 cells, 12,217 run events), 10 (97 cells, 18,505 run events) and 15 (96 cells, 15,928 run events) rounds in rich media and in minimal media for the founding strain (72 cells, 7556 run events), round 5 (45 cells, 9724 run events) and round 10 (25 cells, 4892 run events). (**a**) Shows the fraction of runs longer than a given $\tau_r$ for strains evolved in rich media (95% confidence intervals from bootstrapping). The mean and standard deviation in run duration for founder is 0.66 ± 0.78 s, for round 5: 0.63 ± 0.61 s, for round 10: 0.58 ± 0.50 s and for round 15: 0.65 ± 0.57 s. Round 5, 10 and 15 strains exhibit shorter average run durations than founder ($p<0.05$). (**b**) Shows the same distribution for strains in minimal medium with founder exhibiting average run duration 0.49 ± 0.52 s, round 5: 0.44 ± 0.48 s and round 10: 0.33 ± 0.28 s. Rounds 5 and 10 exhibit shorter average run durations than founder ($p<10^{-8}$). (**c–d**) Show average fraction of time spent tumbling (tumble bias) and tumble frequency (tumbles per second) for rich medium and minimal medium respectively. Note the two vertical axes. In rich medium only the round 15 tumble bias is significantly different from founder ($p<0.001$), but the tumble frequency is higher than founder for both rounds 10 and 15 ($p<0.001$). In minimal medium all tumble biases and frequencies are significantly different from founder for all strains ($p<0.001$). (**e**) Shows run speed distributions for strains evolved in rich medium, legend in (**a**) applies. The average ± standard deviation run speeds are, for founder: 18.7 ± 7.1 μm s$^{-1}$, round 5: 24.9 ± 7.1 μm s$^{-1}$, round 10: 27.6 ± 7.0 μm s$^{-1}$, and for round 15: 28.7 ± 6.8 μm s$^{-1}$. Average run speeds for rounds 5, 10 and 15 are greater than founder (**f**) Shows the same distributions for strains evolved in minimal medium, average run speed for founder: 20.7 ± 10.8 μm s$^{-1}$, for round 5: 11.2 ± 4.8 μm s$^{-1}$ and for round 10: 13.3 ± 4.4 μm s$^{-1}$. Both rounds 5 and 10 exhibit slower average run speeds than founder, the legend in (**b**) applies. (**g–h**) Show growth rates in well mixed liquid culture for all strains studied in panels (**a–f**) in the medium in which the strains were selected. (**g**) Shows triplicate measurements from each of the four strains isolated in rich medium. Rounds 5, 10 and 15 exhibit slower growth than founder ($p<0.01$). (**h**) Shows growth rates for strains isolated from minimal medium selection experiment. Four replicate measurements were made for founder and round 10 and three replicate measurements for round 5. Squares and circles demarcate measurements made on separate days. Rounds 5 and 10 have higher growth rates than founder ($p<10^{-5}$).

The following figure supplements are available for figure 3:

**Figure supplement 1.** Microfluidic device and single-cell swimming trajectory.

**Figure supplement 2.** Tumble durations and run lengths for evolved strains.

*Figure 3 continued on next page*

*Figure 3 continued*

**Figure supplement 3.** Reproducibility of the evolved phenotype.

**Figure supplement 4.** Swimming statistics as a function of culture density.

Combining growth rate measurements with single-cell motility measurements allowed us to predict the front migration rate for strains in rich and minimal media using the reaction-diffusion model described above. We found that the model qualitatively recapitulated the increase in front migration rate that we observed experimentally (*Tables 3*,*4*, *Figure 4—figure supplement 1*).

We conclude that there is a trade-off between run speed and growth rate in *E. coli* which constrains the evolution of faster migration through low viscosity agar. *Figure 4*, which summarizes this trade-off for both conditions, shows the measured growth rates and swimming speeds for all strains presented in *Figure 3* overlaid on the predicted migration rates from our reaction-diffusion model. The curves in *Figure 4a–b* show that the evolved phenotypes lie near a Pareto frontier in the phenotypic space of run speed and growth rate.

## Parallel genomic evolution drives a trade-off through antagonistic pleiotropy

To investigate the mechanism of the phenotypic evolution and trade-off we observed, we performed whole genome sequencing of populations for the founding strain as well as strains isolated after rounds 5, 10 and 15 in rich medium for four of five selection experiments and rounds 5 and 10 in minimal medium for four of five selection experiments (Materials and methods). *Figure 5* shows de novo mutations observed in each strain sequenced. Since we sequenced populations, we report the frequency of each mutation observed (see legend, *Figure 5a*, middle panel).

In the rich medium experiment we observed parallel evolution across replicate selection experiments, with a mutation in *clpX* (E185*) and an intergenic single base pair deletion both rising to

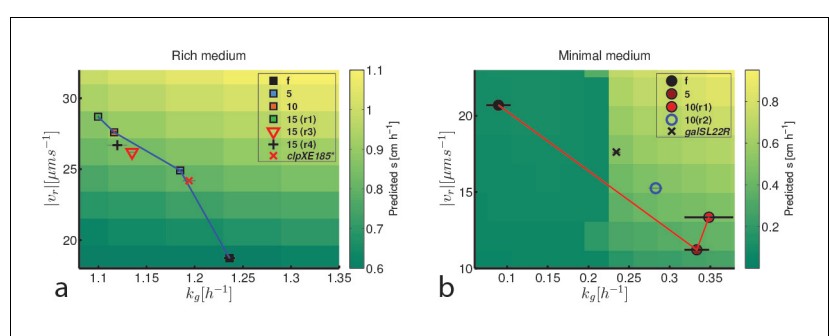

**Figure 4.** Trade-off between growth rate and run speed constrains evolution of faster migration. (a) Shows run speeds and growth rates for strains evolving faster migration in rich medium overlaid on a heatmap of the prediction for front migration rate from the reaction-diffusion model (*Figure 2*). Phenotypes for strains from *Figure 3* are shown along with two independently evolved strains (replicates 3 (15(r3)) and 4 (15(r4)), *Figure 1c*). In addition, the red 'x' marks the phenotype for the mutation *clpX*E185* in the founding strain background (*Figure 5*). (b) Shows run speeds and growth rates for strains evolved in minimal medium overlaid on the predicted from migration rate from the reaction-diffusion model. Growth rate and run speed for an independently evolved round 10 strain is shown (10(r2), *Figure 1e*) as well as the phenotype for the *galS*L22R mutation in the founder background (black 'x'). Predicted front migration rates assume no change in run duration.

The following figure supplements are available for figure 4:

**Figure supplement 1.** Predicted migration rates for evolved strains.

**Figure supplement 2.** Swimming statistics, growth rates and migration rates for mutants.

fixation within approximately 5 rounds of selection. In this condition we observed transient mutations in genes regulating chemotaxis or motility (near *flhD*, *Figure 5a*) in two of four replicates.

A previous study showed that mutations in *clpX* alter *flhDC* expression and motility (*Girgis et al., 2007*). We therefore focused attention on the mutation in *clpX*, which converted position 185 from glutamic acid to a stop codon in the 424 residue *ClpX* protein. *ClpX* is the specificity subunit of the *ClpX-ClpP* serine protease. *ClpX* forms a homohexamer that consumes ATP to unfold and translocate target proteins to the *ClpP* peptidase (*Baker and Sauer, 2012*). The *ClpXP* protease has many targets in the cell including *FlhDC*, the master regulator of flagellar biosynthesis (*Tomoyasu et al., 2003*). We found that this mutation in *clpX* was at high abundance (>70%) in all populations after 5 rounds of selection and fixed by round 10 in all four replicates (*Figure 5a*).

To determine the phenotypic effects of *clpX*E185*, we used scarless recombineering to reconstruct this mutation in founding strain genetic background (*Kuhlman and Cox, 2010*) (Materials and methods). We then performed migration rate, single-cell tracking and growth rate measurements on this strain. We observed a statistically significant increase in migration speed for the *clpX*E185* mutant ($0.39 \pm 0.01$ cm h$^{-1}$, mean and standard error) relative to founder ($0.30 \pm 0.01$ cm h$^{-1}$, $p = 0.002$). We also found that *clpX*E185* resulted in a statistically significant increase in run speed relative to founder (24.2 µm s$^{-1}$ compared to 18.7 µm s$^{-1}$, p< 10$^{-10}$). Finally, in well mixed batch culture in rich medium, the *clpX*E185* mutant exhibited a maximum growth rate $k_g = 1.19 \pm 0.009$ h$^{-1}$ (standard error for triplicate measurements) with founder exhibiting a maximum growth rate of $1.23 \pm 0.01$ h$^{-1}$ ($p = 0.0174$, *Figure 4—figure supplement 2*). Knocking out *clpX* from founder resulted in very slow front migration ($s = 0.0036 \pm 0.001$ cm h$^{-1}$), suggesting that the stop codon mutation we observe has a more subtle effect on the enzyme's function than a simple loss of function. Finally, we reconstructed the intergenic single base pair deletion which fixed in

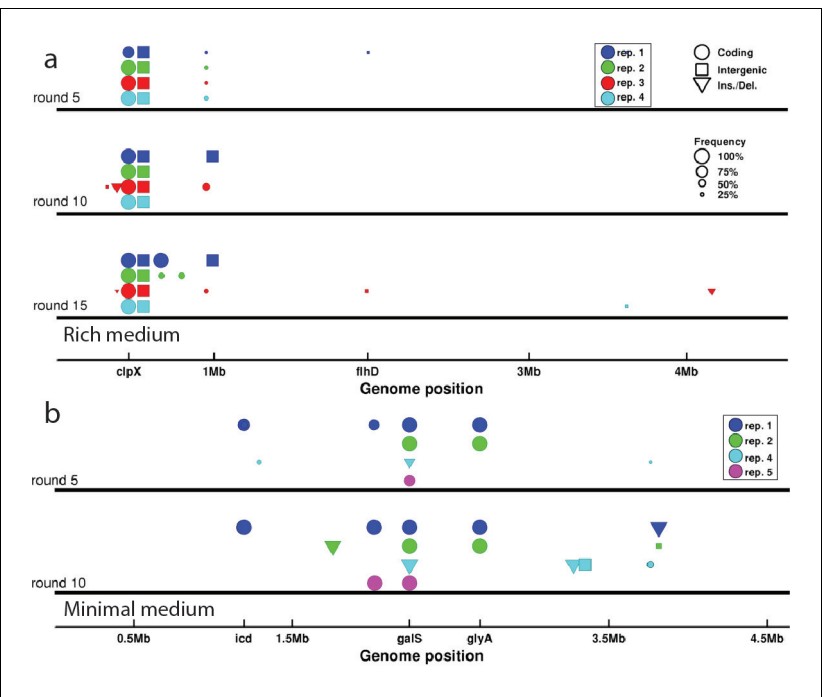

**Figure 5.** Genomic evolution. (a) De novo mutations observed in strains isolated after 5, 10 and 15 rounds of selection in rich medium. Abscissa denotes position along the genome. Colors of the markers indicate independently evolved replicates and correspond to traces in *Figure 1c*. Circles denote single nucleotide polymorphisms (SNP) in coding regions, squares denote intergenic SNPs, and triangles denote larger insertions or deletions. The size of the marker is proportional to the frequency of the mutation in the population. Only mutations with a frequency above 0.2 in the population are shown. Genes of interest are labeled. The operons coding for motility and chemotaxis are near *flhD*. (b) Identical to (a) but shows de novo mutations for strains evolved in minimal medium. The marker near *icd* corresponds to multiple SNPs in close proximity to each other. See *Tables 5–12* for a list of all mutations observed and details of the sequencing.

all four replicate selection experiments but observed no phenotypic effects of this mutation when placed in the founder or *clpX*E185* background (*Figure 4—figure supplement 2*). These results suggest that this intergenic mutation is neutral.

We conclude that the *clpX* mutation observed in all four replicate experiments drives faster front migration through increasing run speed, despite decreasing growth rate. Since the mutant exhibits both faster swimming and slower growth rate relative to founder we conclude that the trade-off between growth rate and swimming speed is driven by antagonistic pleiotropy (*Cooper and Lenski, 2000*).

*Figure 5b* shows the mutations observed in rounds 5 and 10 for four of five replicate selection experiments in minimal medium. In all experiments, we observed mutations in the transcriptional regulator *galS* which fixed in just five rounds. In one of four experiments, we observed a mutation in the gene encoding the motor protein *FliG*, otherwise the observed mutations appear to be metabolic in nature. In minimal medium we also observed a substantial number of synonymous mutations rising to fixation (see *Table 5–8*). The role of these synonymous mutations is not known, but may be due to tRNA pool matching (*Stoletzki and Eyre-Walker, 2007*).

To understand how these mutations drive phenotypic evolution, we focused on the *galS*L22R mutation. *galS* encodes the transcriptional repressor of the *gal* regulon. The coding mutation we observe occurs in the highly conserved N-terminal helix-turn-helix DNA binding region of this protein, we therefore expect that this mutation alters the expression of the *gal* regulon (*Weickert and Adhya, 1992*). To assay the phenotypic effects of this mutation, we reconstructed it in the genetic background of the founder.

The migration rate of the *galS*L22R mutant showed a statistically significant increase relative to founder ($s = 0.039 \pm 0.001$ cm h$^{-1}$ for *galS*L22R and $0.0163 \pm 0.0038$ cm h$^{-1}$ for founder, p < 10$^{-3}$). We found that the growth rate of the mutant was approximately 2.5-fold larger than founder in minimal medium ($0.23 \pm 0.005$ h$^{-1}$ for *galS*L22R and $0.089 \pm 0.03$ h$^{-1}$ for founder, $p = 4 \times < 10^{-4}$). Further, this mutation reduced the mean swimming speed relative to founder by approximately 15% (*Figure 4b*, *Figure 4—figure supplement 2*). However, when we knock out the *galS* gene from founder we do not observe a significant increase in the migration rate ($\Delta galS$ $s = 0.0165 \pm 0.002$ cm h$^{-1}$, $p = 0.92$).

Therefore, as shown in *Figure 4b*, we conclude that *galS*L22R alone drives faster growth and slower swimming. As with the rich medium condition, this trade-off is governed by antagonistic pleiotropy.

## Genetic covariance determines direction of phenotypic evolution

To understand why we observe divergent phenotypic trajectories in the rich and minimal medium conditions (*Figure 4a–b*), we studied a simple model of the evolution of correlated traits (*Lande, 1979*; *Mezey and Houle, 2005*). We consider a vector of the two phenotypes of interest, run speed and maximum growth rate, normalized to the values of the founder (*Hansen and Houle, 2008*), $\vec{\phi} = [|\tilde{v}_r|, \tilde{k}_g]^T$ ($|\tilde{v}_r| = \langle |v_r| \rangle / \langle |v_r|^f \rangle$, $\tilde{k}_g = \langle k_g \rangle / \langle k_g^f \rangle$, where $\langle \rangle$ denotes an average across the population). The model describes the evolution of the mean phenotype ($\vec{\phi}$) under selection by

$$\vec{\phi} = G\vec{\beta} + \vec{\phi}_0 \tag{3}$$

where $G$, the genetic covariance matrix, describes the genetically driven phenotypic covariation in the population, which is assumed to be normally distributed ($\mathcal{N}(\vec{\phi}, G)$). $\vec{\beta}$ is the selection gradient which captures the change in migration rate with respect to phenotype since we are selecting for faster migration. The matrix $G$ is given by

$$G = \begin{bmatrix} \sigma^2_{|\tilde{v}_r|} & \rho\sigma_{|\tilde{v}_r|}\sigma_{\tilde{k}_g} \\ \rho\sigma_{|\tilde{v}_r|}\sigma_{\tilde{k}_g} & \sigma^2_{\tilde{k}_g} \end{bmatrix}, \tag{4}$$

where $\sigma^2_*$ describes the (fractional) variance in the phenotype due to genetic variation and captures the correlation between the two traits. Therefore, the diagonal elements of $G$ describe the capacity for mutations to vary each trait while the off-diagonal elements describe the capacity for mutations to vary both traits. In our experiment we do not have a direct measurement of $G$. However, we do

observe how $\vec{\phi}$ changes over the course of selection, our data suggest that $\rho < 0$ and our reaction-diffusion model permits us to estimate how migration rate depends on the two traits of interest. In particular, $\vec{\beta} = \left[\frac{\partial \log(s)}{\partial(|v_r|)}, \frac{\partial \log(s)}{\partial k_g}\right]^T$.

We approximate $\vec{\beta}$ in both rich and minimal media by fitting a plane to the heatmap shown in *Figure 2a–b* (Appendix 1). The resulting selection gradient is shown in *Figure 6—figure supplement 1* for both conditions. Using this formalism, we asked what values of $\sigma_{\tilde{k}_g}$ and $\sigma_{|\tilde{v}_r|}$ would result in the directions of phenotypic evolution we observed experimentally in rich and minimal media.

We found that the direction of phenotypic evolution in rich medium agreed well with our experimental observations so long as $\sigma_{|\tilde{v}_r|}/\sigma_{\tilde{k}_g} \geq 1$ for $\rho < -0.1$. This implies that our observed phenotypic evolution is consistent with a genetic variance in run speed that is no smaller than the genetic variance in growth rate (*Figure 6—figure supplement 2*). In contrast, in minimal medium the model predicts the direction of observed phenotypic evolution only if $\sigma_{|\tilde{v}_r|}/\sigma_{\tilde{k}_g} \leq 0.3$ for $\rho < -0.1$. This result indicates that our observed phenotypic evolution is consistent with at least three-fold larger propensity for mutations to alter growth rate compared to run speed in minimal medium (*Figure 6—figure supplement 2*). *Figure 6* shows these geometric relationships between selection, genetic covariance and phenotypic evolution.

This suggests that the capacity of mutations to alter run speed or growth rate relative to founder depends on the nutrient conditions and that changes in this capacity qualitatively alter the direction of evolution along a Pareto frontier (*Shoval et al., 2012*). This result captures the intuition that mutations that can increase growth rate in rich medium are few while in minimal medium the propensity for mutations increasing growth rate is substantially larger. The model presented here relies on a linear approximation to $\vec{\beta}$, which is a good assumption for rich medium but not for minimal medium, where the dependence of $s$ on $|v_r|$ and $k_g$ is strongly nonlinear. Using simulations of the evolutionary

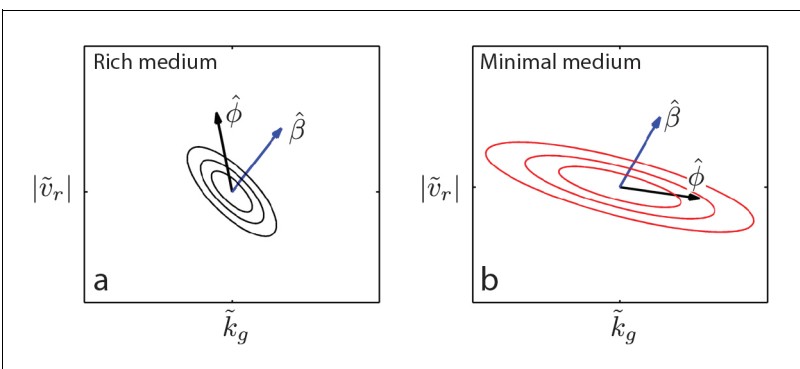

**Figure 6.** Evolution of correlated traits. The evolutionary model describes the change in phenotype relative to the founder ($\vec{\phi} = [|\tilde{v}_r|, \tilde{k}_g]^T$) under selection described by $\vec{\beta}$. Panels show unit vectors in the direction of observed phenotypic evolution ($\hat{\phi}$) and the direction of selection inferred from the reaction-diffusion model ($\hat{\beta}$). Ellipses show quartiles for a normal distribution of phenotypes with covariance matrix $G$ that is consistent with $\vec{\phi}$ and $\vec{\beta}$. In both panels, we set the correlation coefficient between $\tilde{k}_g$ and $|\tilde{v}_r|$ is $\rho = -0.75$ but our conclusions hold for $\rho < -0.1$. In rich medium (a) $\sigma_{|\tilde{v}_r|}/\sigma_{\tilde{k}_g} = 1$ and in minimal medium $\sigma_{|\tilde{v}_r|}/\sigma_{\tilde{k}_g} = 0.3$. In rich medium $\hat{\beta}_{RM} = [0.78, 0.61]$ and in minimal medium $\hat{\beta}_{MM} = [0.87, 0.49]$.

The following figure supplements are available for figure 6:

**Figure supplement 1.** Determining $\vec{\beta}$ from reaction-diffusion model.

**Figure supplement 2.** Direction of phenotypic evolution with $\sigma_{|\tilde{v}_r|}$ and $\sigma_{\tilde{k}_g}$.

**Figure supplement 3.** Stochastic simulations of selection in minimal medium.

process described by *Equation 3* , we relaxed the assumption of linearity in the selection coefficient and found that our qualitative conclusions were not altered (*Figure 6—figure supplement 3*).

We note that the structure of $G$ inferred above reflects the capacity for mutations to change phenotypes at the outset of the experiment. As evolution proceeds in rich medium, we observe a saturation in both run speed and growth rate (*Figure 4a*), suggesting that further variation is constrained, either genetically or through biophysical constraints on swimming speed. Similarly, in minimal medium, saturation in the growth rate occurs after 5 rounds of selection, suggesting that mutations to further improve growth rate are either not available or fundamental constraints on growth inhibit further increases (*Scott et al., 2010*).

## Discussion

The most striking observation of our study is the divergent trajectories of phenotypic evolution shown in *Figure 4a–b*. This observation shows that the evolution of faster migration results in environmentally dependent phenotypic outcomes. This result has important implications for interpreting phenotypic variation in natural populations.

When trade-offs are observed in wild populations, it is sometimes proposed that phenotypes at the extrema of a Pareto frontier reflect the outcome of selection for a specific task (*Shoval et al., 2012*). Our study shows that when selection pressures place demands on multiple traits simultaneously, evolution along the frontier can reflect differing genetic capacity for adaptation of each phenotype rather than simply the fitness benefit of improving each trait. This result suggests a cautious approach to interpreting phenotypes in nature, where selection pressures and mechanisms constraining phenotypes are often not known (*Gould and Lewontin, 1979*).

Our results point to the potential predictive power of determining the directions in phenotype space in which genetic variation can most readily change phenotypes – so called, 'genetic lines of least resistance' (*Schluter, 1996*). These directions may be related to genetic regulatory architecture. The mutations we observe in both rich and minimal media alter negative regulators (a protease in the case of *clpX* and a transcriptional repressor in the case of *galS*). This supports the hypothesis that microevolution is dominated by the disruption of negative regulation (*Lind et al., 2015*) and suggests that the direction of phenotypic evolution can be predicted by determining where negative regulatory elements reside in genetic and proteomic networks. The mutations we examined appear to be more subtle than simple loss of function, since knockout mutants for both *clpX* and *galS* do not exhibit fast migration, therefore a detailed understanding of how mutations disrupt negative regulation will be essential.

Previous experimental evolution studies have revealed a similar trade-off to the one presented here. Comparing the results of these studies to our own demonstrates the impact of how selection is performed on the phenotypic outcomes. For example, *Yi and Dean, 2016* selected *E. coli* alternately for growth in well mixed conditions and chemotaxis using a capillary assay and observed a trade-off between growth rate and swimming speed which was circumvented by phenotypic plasticity. We observe no evolution beyond the Pareto frontier in our study, possibly because our conditions simultaneously select for growth and motility rather than alternating between selection pressures. This suggests that evolutionarily persistent trade-offs may reflect selection pressures that occur simultaneously in nature. In addition, *van Ditmarsch et al. (2013)* and *Deforet et al. (2014)* select *Pseudomonas aeruginosa* for a hyperswarming phenotype on hard agar. Rather than sampling from the population at a specific location in a swarming colony, they allow the population to swarm for a fixed time interval, remove the entire colony from the plate and inoculate a second plate from a mixed sample of the entire colony. This procedure likely selects both for swarming speed and for growth in the bulk of the colony. Phenotypically, hyperswarmers selected in this way exhibit a decline in growth rate and swimming speed in liquid and a deficit in biofilm formation (*van Ditmarsch et al., 2013*; *Deforet et al., 2014*). In light of our study, these results suggest that evolved phenotypes can depend on whether selection occurs at well defined spatial locations in a structured population (e.g. migrating fronts) or through periodic removal of spatial structure. A more precise understanding of the selection pressure applied by van Ditmarsch *et al.* might emerge from the application of Lande's (*Lande, 1979*) formalism to the observed genetic and phenotypic variation.

Interestingly, both *Yi and Dean (2016)* and *van Ditmarsch et al. (2013)* observe mutations that alter regulation of motility and chemotaxis genes. None of the mutations observed in our experiment were found by Yi and Dean, despite evolution along similar Pareto frontiers. This suggests that determining the allowed directions of phenotypic variation may be a more powerful approach to predicting evolution than cataloging mutations alone.

The mechanism of the trade-off between growth rate and swimming speed has, to our knowledge, not been determined. However, over-expression of motility operons could drive the reductions in growth rate we observe in rich medium. Subsequent increases in speed could then arise passively from reductions in cell size which reduce hydrodynamic drag (*Taheri-Araghi et al., 2015*). Similarly, increases in growth rate in minimal medium should increase cell size and hydrodynamic drag. Using the data of *Taheri-Araghi et al. (2015)*, we estimated changes in cell size due to measured changes in growth rate for populations evolved in rich and minimal medium. We could not account for the large change in swimming speed we observe through growth rate mediated changes in cell size alone (Appendix 1). Since we have not measured cell size directly, we cannot conclusively rule out this mechanism. To definitively characterize the mechanism of this trade-off will require measurements of cell size, gene expression, flagellar length and proton motive force.

Our study shows how evolutionary dynamics are defined by the complex interplay between genetic architecture, phenotypic constraints and the environment. Our hope is that a general approach to predicting evolution can emerge from a more complete understanding of this interplay.

## Materials and methods

### Motility selection

#### Rich medium

10 µL of motile *E. coli* (strain MG1655-motile, Coli Genetic Stock Center (CGSC) #8237) from an overnight LB culture was injected at the center of a 0.3 % w/v agar 15 cm diameter plate containing LB. Images were acquired every minute via webcams (Logitech HD Pro Webcam C920, Logitech, Lausanne, Switzerland) in a dark box using pulsed illumination provided by an LED light strip (part number: COM-12021, Sparkfun Inc, Nilwot, CO). Only the Red and Green LEDs of the RGB strip were used in order to avoid blue light response known to occur in *E. coli* (*Taylor and Koshland, 1975*). After 12 hr, 50 µL of cells was removed from each of eight points around the outermost ring. This sample was briefly vortexed and 10 µL ($\approx 10^6$ cells) was injected into a fresh plate from the same batch. Remaining bacteria were preserved at $-80°C$ in 25% glycerol. Selection was performed by repeating this sampling and growth over 15 rounds. Automated image processing then yielded quantitative data about front speed. All experiments were performed at 30°C in an environmental chamber (Darwin Chambers, St. Louis, MO). Plates were allowed to equilibrate for 12 hr at 30°C in the environmental chamber before use. All plates for a single selection replicate were poured from a single media bottle. Plates were allowed to cool, parafilmed and stored at 4°C until use.

To estimate the number of generations that occur during each round of selection, we inoculated an agar plate from a culture of the founding strain. We then measured the cell density of the inoculum by serial dilution and plating. We permitted the colony to expand for 12 hr. To measure the total population on the plate after growth, we mixed the entire contents of the plate in a beaker and measured the density by serial dilution and plating again. From this we extracted an estimate of the number of generations that occurred. The range reflects errors due to serial dilution and plating and the difference in colony size during selection.

#### Minimal medium

Selection experiment was performed identically to rich medium experiment with the following modifications. Plates were made with M63 0.18 mM galactose. Cultures used to initiate selection were grown in M63 30 mM galactose for 24 to 48 hr prior to initiating selection. During each 48 hr round of migration and imaging, plates were housed in a plexiglass box with a beaker of water to prevent evaporative losses from the plate. Images were acquired every 2 min. We estimated the number of generations per round as described above. Reliable plate counts were only obtained for plates of round 10 strains where we estimate 10 generations per round. We therefore take this as an upper

**Table 1.** Reaction-diffusion model parameters: Columns indicate parameter, explanation of parameter, units, value used in simulation of founder strain in rich medium, and the value used in simulation of founder strain in minimal medium. Parameters marked with an $^m$ were measured in this study. $D_b$, $k_0$ and $D_c$ in rich medium were estimated as described in Appendix 1 using the methods of **Croze et al (2011)**. $D_c$ is assumed to be the same in minimal medium as rich medium. Identical $k_0$ and $D_b$ were used in rich and minimal media since **Ford and Lauffenburger (1992)** find nearly identical values for galactose as **Ahmed and Stocker (2008)** do for serine. $K_D$ for both nutrient conditions was taken from Hazelbauer **Adler et al., 1973**. For minimal medium $K_g$ and $Y$ were taken from **Lendenmann et al., 1999**. The values cited for $s$ were measured from numerical simulation of the reaction-diffusion model as outlined in Materials and methods.

| Parameter | Explanation | Units | Founder value RM | Founder value MM |
|---|---|---|---|---|
| Single-cell swimming (this study) | | | | |
| $\tau_r$ | run duration | s | $0.67^m$ | $0.47^m$ |
| $\tau_t$ | tumble duration | s | – | – |
| $|v_r|$ | run speed | $\mu m\ s^{-1}$ | $18.7^m$ | $22.2^m$ |
| Reaction-diffusion model | | | | |
| $\rho(\mathbf{r},t)$ | cell density | $m^{-3}$ | – | – |
| $c(\mathbf{r},t)$ | nutrient density | mM | – | – |
| $c_0$ | nutrient concentration in medium | mM | $1^m$ | $0.18^m$ |
| $D_b$ | bacterial diffusion constant | $cm^2h^{-1}$ | 0.0576 | 0.0576 |
| $D_c$ | nutrient diffusion constant | $cm^2h^{-1}$ | 0.036 | 0.036 |
| $k_0$ | chemotactic coefficient in liquid | $cm^2h^{-1}$ | 6.12 | 6.12 |
| $K_D$ | receptor-nutrient binding constant | mM | 2 | 0.1 |
| $k_g$ | maximum growth rate | $h^{-1}$ | $1.23^m$ | $0.125^m$ |
| $K_g$ | $c$ concentration for half-maximum growth rate | mM | 0.1 | 3P $<10^{-4}$ |
| $Y$ | yield biomass per unit nutrients | cells $mL^{-1}mM^{-1}$ | $5\times10^{7m}$ | $3\times10^8$ |
| $C$ | agar concentration | % (w/v) | $0.3^m$ | $0.3^m$ |
| $s$ | front migration rate | cm $h^{-1}$ | 0.61 | 0.09 |

bound and conclude that the 10 round selection experiment includes <100 generations. Plates were thermalized for 24 hr before use.

The $\Delta cheA$-$Z$ mutant was constructed via P1 transduction from a strain provided by the group of Chris Rao and the mutation was confirmed by PCR. This mutant lacks the receptors *tar* and *tap* and the chemotaxis genes *cheAWRBYZ*.

**Table 2.** Tumble bias and frequency for additional strains.

**Tumble bias and frequencies**

| strain | Tumble bias | Tumble frequency [Hz] |
|---|---|---|
| Rich medium | | |
| founder | $0.197 \pm 0.006$ | $0.59 \pm 0.02$ |
| 15(r3) | $0.174 \pm 0.006$ | $0.78 \pm 0.02$ |
| 15(r4) | $0.2 \pm 0.01$ | $0.79 \pm 0.01$ |
| *clpX*E185* | $0.19 \pm 0.01$ | $0.66 \pm 0.02$ |
| Minimal medium | | |
| founder | $0.29 \pm 0.01$ | $0.41 \pm 0.03$ |
| 10(r2) | $0.25 \pm 0.02$ | $0.44 \pm 0.03$ |
| *gal*SL22R | $0.3 \pm 0.02$ | $0.44 \pm 0.05$ |

**Table 3.** Reaction-diffusion model parameters estimated from measurements of tumble frequency ($\alpha_0$) and run speed ($|v_r|$) for rich medium evolved strains in $C = 0.3\%$ agar.

**Evolution of population level migration parameters**

| strain | $\alpha_0$ [s$^{-1}$] | $|v_r|$ [ μm s$^{-1}$] | $D_b$ [ cm$^2$h$^{-1}$] | $k_0$ [ cm$^2$h$^{-1}$] |
|---|---|---|---|---|
| founder | 1.45 | 18.7 | 0.02 | 0.65 |
| round 5 | 1.56 | 24.9 | 0.027 | 0.90 |
| round 10 | 1.72 | 27.6 | 0.029 | 1.04 |
| round 15 | 1.54 | 28.7 | 0.031 | 1.04 |

We selected the motile MG1655 wild-type strain for these experiments rather than the more commonly used RP437 strain since the latter is auxotrophic for several amino acids. Minimal medium experiments were therefore performed without additional amino acids which could confound results.

## Image analysis

Webcam acquired images of migrating fronts were analyzed by custom written software (Matlab, Mathworks, Natick, MA). A background image was constructed by median projecting six images from the beginning of the acquisition before significant growth had occurred. This image was subtracted from all subsequent images prior to further analysis. The location of the center of the colony was determined by first finding the edges of the colony using a Canny edge detection algorithm. A circular Hough transform (*Hough, 1959*) was applied to the resulting binary image to locate the center. In rich medium, where signal to background was >10, radial profiles of image intensity were measured from this center location and were not averaged azimuthally due to small departures from circularity in the colony. The location of the front was determined by finding the outermost peak in radial intensity profiles. Migration rate was determined by linear regression on the front location in time. Imaging was calibrated by imaging a test target to determine the number of pixels per centimeter. The results of the calibration did not depend on the location of the test target in the field of view. In minimal medium, where the signal to background is reduced due to low cell densities, background subtraction was employed as described above but radial density profiles were not always reliable for locating the front. Instead, a circular Hough transform was applied to each image to locate the front at each point in time.

## Single-cell tracking

Single-layer microfluidic devices were constructed from polydimethyl-siloxane (PDMS) using standard soft-lithography techniques, (*Quake and Scherer, 2000*) following a design similar to the one used previously (*Jordan et al., 2013*), and were bonded to coverslips by oxygen plasma treatment (Harrick plasma bonder, Harrick Plasma, Ithaca, NY). Bonded devices formed a circular chamber of diameter 200 μm and depth 10 μm (*Figure 3—figure supplement 1*). Devices were soaked in the medium used for tracking (LB for rich medium strains, M63 0.18 mM galactose for minimal medium strains) with 1% Bovine Serum Albumin (BSA) for at least 1 hr before cells were loaded. Bacteria were inoculated directly from frozen stocks into medium containing 0.1% BSA in a custom continuous culture device. BSA was necessary to minimize cells adhering to the glass cover slip. For rich

**Table 4.** Reaction-diffusion model parameters estimated from measurements of tumble frequency ($\alpha_0$) and run speed ($|v_r|$) for minimal medium evolved strains in $C =0.3\%$ agar.

**Evolution of population level migration parameters**

| strain | $\alpha_0$ [s$^{-1}$] | $|v_r|$ [ μm s$^{-1}$] | $D_b$ [ cm$^2$h$^{-1}$] | $k_0$ [ cm$^2$h$^{-1}$] |
|---|---|---|---|---|
| founder | 2 | 20.7 | 0.021 | 0.66 |
| round 5 | 2.5 | 11.2 | 0.011 | 0.39 |
| round 10 | 3 | 13.3 | 0.011 | 0.5 |

**Table 5.** Minimal medium replicate 1: All mutations detected in rounds 5 and 10 of minimal medium replicate 1. The *galS*L22R mutation in rounds 5 and 10 was confirmed by Sanger sequencing. See *Table 9* caption.

**Minimal medium replicate 1**

| round, (coverage) | 5, (71×) | 10, (214×) |
|---|---|---|
| **Mutation (loc., mut., frac., cov.)** | 1196220, *icd* H366H, 78.4%, 46 | 1196220, *icd* H366H, 100%, 141 |
| | 1196232, *icd* T370T, 71.1%, 34 | 1196232, *icd* T370T, 100%, 101 |
| | 1196247, *icd* L375L, 72.0%, 25 | 1196247, *icd* L375L, 100%, 75 |
| | 1196277, *icd* N385N, 47.1%, 17 | 2015871, *fliG* V331D, 100%, 111 |
| | 1196280, *icd* A386A, 47.1%, 17 | 2241604, *galS* L22R, 100%, 184 |
| | 1196283, *icd* K387K, 47.2%, 17 | 2685013, *glyA* H165H, 100%, 197 |
| | 1196292, *icd* T390T, 46.2%, 13 | 3815859, *rph* Δ82 bp, 100%, 260 |
| | 1196304, *icd* E394E, 46.2%, 13 | |
| | 2015871, *fliG* V331D, 70.0%, 60 | |
| | 2241604, *galS* L22R, 100%, 45 | |
| | 2685013, *glyA* H165H, 100%, 62 | |

medium tracking experiments, cells grew to a target optical density and the continuous culture device was run as a turbidostat. In minimal medium experiments the device was run as a chemostat at an optical density of ~0.15. The culture was stirred by a magnetic stir bar at 775 RPM and the temperature was maintained at 29.75°C by feedback.

To perform single-cell tracking, cells were sampled from the continuous-culture device and diluted appropriately (to trap one cell in the chamber at a time) before being pumped into the microfluidic chamber. Video was acquired at 30 frames per second with a Point Grey model FL3-U3-32S2M-CS camera (Point Grey, Richmond, Canada) and a bright-field microscope (Omano OM900-T inverted) at 20x magnification. Movies were recorded for 5 min before a new cell was loaded into the chamber. An example movie is available at https://doi.org/10.13012/B2IDB-4912922_V2. Two microscopes were operated in parallel. The stock microscope light source was replaced by a high-brightness white LED (07040-PW740-L, LED Supply, Randolf, VT) to avoid 60 Hz flickering that was observed with the stock halogen light source. All experiments were performed in an environmental chamber maintained at 30°C.

Movies were segmented and tracked with custom written Matlab routines described previously (*Jordan et al., 2013*; *Jaqaman et al., 2008*). Code is available at https://github.com/dfraebel/Cell-Tracking (*Mickalide et al., 2017*; copy archived at https://github.com/elifesciences-publications/CellTracking). At times when two individuals are present in the chamber, ambiguous crossing events can lead to loss of individual identities. All crossing events were inspected manually to prevent this. To identify runs and tumbles, we utilized a method based on reference (*Taute et al., 2015*) which was modified from the approach used by *Berg and Brown (1972)*. Briefly, for each cell the segmentation routine results in a matrix of spatial locations $\vec{x}(t)$. We compute the velocity by the method of

**Table 6.** Minimal medium replicate 2: All mutations detected in rounds 5 and 10 of minimal medium replicate 2. The *galS*L22R mutation in rounds 5 and 10 was confirmed by Sanger sequencing. See *Table 9* caption.

**Minimal medium replicate 2**

| round, (coverage) | 5, (67×) | 10, (64×) |
|---|---|---|
| | 2241604, *galS* L22R, 100%, 70 | 1757419, IG +17 bp insertion, 94.9%, 37 |
| **Mutation (loc., mut., frac., cov.)** | 2685013, *glyA* H165H, 100%, 65 | 2241604, *galS* L22R, 100%, 47 |
| | | 2685013, *glyA* H165H, 100%, 79 |
| | | 3815828, IG T→G, 43.5%, 62 |

**Table 7.** Minimal medium replicate 3: All mutations detected in rounds 5 and 10 of minimal medium replicate 3. See *Table 9* caption.

| Minimal medium replicate 3 | | |
|---|---|---|
| round, (coverage) | 5, (208×) | 10, (229×) |
| | 1291079, *rssB* A280T, 29.7%, 54 | 2241595, *galS* Δ1bp, 100%, 218 |
| Mutation (loc., mut., frac., cov.) | 2241595, *galS* Δ1bp, 64.7%, 102 | 3277264, *prlF* +CATTCAA insertion, 93.6%, 109 |
| | 3762200, *rhsA* A6A, 23.5%, 181 | 3350529, IG T→C, 100%, 117 |
| | 3762212, *rhsA* G10G, 23.1%, 164 | 3762200, *rhsA* A6A, 45.8%, 320 |
| | | 3762212, *rhsA* G10G, 42.0%, 292 |

central differences resulting in $\vec{v}(t)$ from which we compute an angular velocity between adjacent velocity vectors ($\omega(t)$). We then define $\alpha$, a threshold on $\omega$. Tumbles are initiated if $\omega(t)$ & $\omega(t+1) > \alpha$ or if $\omega(t) > \alpha$ and the angle defined between the vectors $\vec{x}(t-2) - \vec{x}(t)$ and $\vec{x}(t) - \vec{x}(t+2)$ is greater than $\alpha$. The latter condition detects tumbles that occur on the timescale of the imaging (0.033 s). Runs are initiated only when $\omega(t)$ & $\omega(t+1)$ & $\omega(t+2) < \alpha$. As a result, tumbles can be instantaneous and runs are a minimum of four frames. $\alpha$ was determined dynamically for each individual by initializing $\alpha_0$ and then detecting all runs for a cell. A new $\alpha_i = c \times \mathrm{median}(\omega_{runs})$ was computed with $c$ a constant and $\omega_{runs}$ is the angular velocity during runs. The process was iterated ten times but typically converged to a final $\alpha_f$ in less than five iterations. $c = 5$ was determined by visual inspection of resulting classified trajectories. Approximately, 1% of cells exhibited sustained tumbling and had average tumble durations greater than 0.4 s and were excluded from further analysis.

We only considered run events that were in the bulk of the chamber and were not interrupted by interactions with the circular boundary of the chamber. We computed tumble bias by measuring the total time spent tumbling when the cell was not interacting with the chamber boundaries. Tumble frequency was computed by counting the number of tumble events that occurred in the bulk of the chamber and dividing by the total time the cells spent swimming in the bulk. Tumble bias and frequency were computed for each individual over the duration tracked. Averages across individuals are reported in *Figure 3c–d*.

Due to interactions with the chamber floor and ceiling (boundaries perpendicular to the optical axis), we intermittently observed cells circling. We developed a method to detect this behavior automatically and found that our results are unchanged when we consider individuals that are not interacting with the chamber boundaries (Appendix 1). Data presented in the main text excludes cells determined to be circling.

## Whole genome sequencing and analysis

Whole genome sequencing was performed using the Illumina platform with slight variations between four independent runs. For all sequencing, cultures were grown by inoculating fresh medium from frozen stocks isolated during the course of selection and growing to saturation at 30°C. For sequencing of rich medium strains from replicate 1, DNA was extracted and purified using a Bioo Scientific NEXTprep-Bacteria DNA Isolation Kit. Libraries were prepared from these strains with the Kapa HyperLibrary Preparation kit (Kapa Biosystems, Wilmington MA), pooled and quantified by qPCR and sequenced for 101 cycles from each of the fragments on a HiSeq 2500 (Illumina, San Diego, CA). This HiSeq run was performed by the Biotechnology Core Facility at the University of Illinois at

**Table 8.** Minimal medium replicate 4: All mutations detected in rounds 5 and 10 of minimal medium replicate 4. See *Table 9* caption.

| Minimal medium replicate 4 | | |
|---|---|---|
| round, (coverage) | 5, (256×) | 10, (230×) |
| | 2241232, *galS* R146L, 72.4%, 274 | 2020519, *fliM* E145K, 100%, 205 |
| Mutation (loc., mut., frac., cov.) | | 2241665, *galS* I2L, 100%, 304 |

**Table 9.** Rich medium replicate 1: All mutations detected above a frequency of 0.2 in rounds 5, 10 and 15 of rich medium selection replicate 1. The first number in each cell denotes the distance in base pairs from *ori* (location). The second entry (mutation) identifies the mutations with 'IG' denoting an intergenic mutation. The third entry (fraction) is the fraction of the population carrying this mutation (as inferred by breseq in polymorphism mode). The fourth entry (coverage) is the number of reads that aligned to this location. In the round 15 strain, the *clpX* SNP and Δ1 bp deletion at position 523,086 were confirmed by Sanger sequencing.

**Rich medium replicate 1**

| Round, (coverage) | 5, (172×) | 10, (213×) | 15, (180×) |
|---|---|---|---|
| | 457978, *clpX* E185*, 75.2%, 179 | 457978, *clpX* E185*, 100%, 199 | 457978, *clpX* E185*, 100%, 164 |
| Mutation (loc., mut., frac., cov.) | 523086, IG Δ1 bp, 100%, 194 | 523086, IG Δ1 bp, 100%, 266 | 523086, IG Δ1 bp, 100%, 168 |
| | 950518, *pflA* T188I, 22.2%, 144 | 990379, IG A→C, 100%, 201 | 663115, *dacA* Δ1 bp, 100%, 150 |
| | 1978458, IG G→T, 21.2%, 156 | | 990379, IG A→C, 100%, 156 |
| | 3618863, *nikR* H92H, 20.7%, 189 | | |

Urbana-Champaign and included additional strains not presented here. All other sequencing was performed on a locally operated and maintained Illumina MiSeq system.

For MiSeq runs which generated data for all minimal medium evolved strains and replicates 2 to 4 of the rich medium selection experiments, DNA was extracted with either the Bioo Scientific NEXTprep kit or the MoBio Ultraclean Microbial DNA isolation kit. Different isolation kits were used due to the discontinuation of the Bioo Scientific kit. DNA was quantified by Qubit and Bioanalyzer and libraries were prepared using the NexteraXT kit from Illumina.

Sequencing adapters for the HiSeq generated data were trimmed using *flexbar* (http://sourceforge.net/projects/flexbar/). MiSeq runs were demultiplexed and trimmed using the onboard Illumina software. Analysis was performed using the *breseq* platform http://barricklab.org/twiki/bin/view/Lab/ToolsBacterialGenomeResequencing in polymorphism mode. *Breseq* uses an empirical error model and a Bayesian variant caller to predict polymorphisms at the nucleotide level. The algorithm uses a threshold on the empirical error estimate (E-value) to call variants (*Barrick and Lenski, 2009*). The value for this threshold used here was 0.01, and at this threshold, with the sequencing coverage for our samples, we report all variants present in the population at a frequency of 0.2 or above (*Barrick and Lenski, 2009*). All other parameters were set to their default values. Reads were aligned to the MG1655 genome (INSDC U00096.3). We note that *breseq* is not well suited to predicting large structural variation. Since we sequence populations at different points during selection, observation of the same mutations at different points in time significantly reduces the probability of false positives (*Lang et al., 2013*).

The founder strain was sequenced at an average depth of 553× when aggregating reads from four separate sequencing reactions. Any mutations observed in this strain were excluded from further analysis. *Tables 5–12* document mutations, important mutations were confirmed by Sanger sequencing as noted in the captions to these tables. Since these genomes were sequenced at very high depth, we did not confirm every mutation by Sanger sequencing. All mutation calls made by *breseq* were inspected manually and found to be robust or they were excluded. We also manually inspected the founder strain reads aligned to regions where frequent mutations were observed in

**Table 10.** Rich medium replicate 2: All mutations detected in rounds 5, 10 and 15 of rich medium replicate 2. See **Table 9** caption. Note low coverage on Δ1 bp mutation at 523086 noted in bold.

**Rich medium replicate 2**

| Round, (coverage) | 5, (218×) | 10, (100×) | 15, (166×) |
|---|---|---|---|
| | 457978, *clpX* E185*, 100%, 220 | 457978, *clpX* E185*, 100%, 109 | 457978, *clpX* E185*, 100%, 184 |
| Mutation (loc., mut., frac., cov.) | 950518, *pflA* T188I, 27.2%, 210 | 523086, IG Δ1 bp, 100%, **16** | 523086, IG Δ1 bp, 100%, **24** |
| | 523086, IG Δ1 bp, 100%, **10/18** | | 667259, *mrdA* R320H, 39.5%, 159 |
| | | | 794472, *modE* L58*, 42.4%, 136 |

**Table 11.** Rich medium replicate 3: All mutations detected in rounds 5, 10 and 15 of rich medium replicate 3. See **Table 9** caption. Note low coverage on Δ1 bp mutation at 523086 noted in bold.

**Rich medium replicate 3**

| Round, (coverage) | 5, (291×) | 10, (45×) | 15, (186×) |
|---|---|---|---|
| | 457978, *clpX* E185*, 100%, 300 | 457978, *clpX* E185*, 100%, 43 | 457978, *clpX* E185*, 100%, 185 |
| Mutation (loc., mut., frac., cov.) | 523086, IG Δ1 bp, 50%, **38** | 523086, IG Δ1 bp, 100%, **8** | 523086, IG Δ1 bp, 100%, **16** |
| | 950518, *pflA* T188I, 26.3%, 332 | 950518, *pflA* T188I, 53.3%, 53 | 950518, *pflA* T188I, 30.6%, 190 |
| | | 321263, IG T→C, 25%, 16 | 1968653, *cheR* Q238K, 29.6%, 190 |
| | | 382794, *yaiX* +9bp insertion, 64%, NA | 382794, *yaiX* +9bp insertion, 25.8%, NA |
| | | | 4161562, *fabR* Δ17bp, 46.2%, 67 |

the evolved strains (*clpX* E185*, the Δ1 bp mutation at position 523086 and *galS* L22R) to confirm that those mutations were not present in the founder. Sequencing data are available at https://doi.org/10.13012/B2IDB-3958294_V1.

## Mutant reconstruction

Knockout mutants (Δ*clpX*, Δ*galS*) were constructed by P1 transduction from KEIO collection mutants (**Baba et al., 2006**). Mutations were confirmed by PCR. Antibiotic markers were not removed prior to phenotyping.

Three commonly observed single nucleotide polymorphisms (SNPs) observed across evolution experiments were reconstructed in the chromosome of the ancestral background (founder) using a recombineering method presented previously (**Kuhlman and Cox, 2010**; **Tas et al., 2015**). These mutations were the *clpX*E185* mutation, the single base pair deletion between *ybbP* and *rhsD* (which we refer to as 'Δ1bp') and *galS*L22R. For full details of the recombineering we performed see Appendix 1. Briefly, recombineering proficient cells were prepared by electroporation of the helper plasmid pTKRED (**Kuhlman and Cox, 2010**) and selection on spectinomycin. A linear 'landing pad' fragment consisting of *tetA* flanked by I-SceI restriction sites and homologies to the desired target site was synthesized from the template plasmid pTKLP-*tetA* and site specific primers. The landing pad was inserted by electroporation into recombineering proficient cells and transformants were selected by growth on tetracycline. Successful transformants were confirmed by PCR. A second transformation was then performed using a 70 bp oligo containing the desired mutation near the center and flanked by homologies to target the landing pad. Counterselection for successful transformants was performed with NiCl₂ (6 mM for the *ClpX* and *GalS* mutations, 6.5 mM for Δ1 bp). Successful recombination at this step resulted in removal of the landing pad and integration of the 70 bp oligo containing the desired mutation. The helper plasmid pTKRED was cured by growth at 42°C and confirmed by verifying spectinomycin susceptibility. The presence of desired mutations in the final constructs was confirmed by Sanger sequencing.

**Table 12.** Rich medium replicate 4: All mutations detected in rounds 5, 10 and 15 of rich medium replicate 4. See **Table 9** caption. Note low coverage on Δ1 bp mutation at 523086 noted in bold.

**Rich medium replicate 4**

| Round, (coverage) | 5, (384×) | 10, (555×) | 15, (333×) |
|---|---|---|---|
| | 457978, *clpX* E185*, 100%, 370 | 457978, *clpX* E185*, 100%, 559 | 457978, *clpX* E185*, 100%, 339 |
| | 523086, IG Δ1 bp, 50%, **72** | 523086, IG Δ1 bp, 100%, **34/83** | 523086, IG Δ1 bp, 100%, **19/33** |
| Mutation (loc., mut., frac., cov.) | 950518, *pflA* T188I, 31.7%, 446 | | 3619915, *rhsB* W242G, 24.9%, 20 |

## Acknowledgements

We would like to acknowledge Christopher Rao for strains; Yann Chemla, Andrew Ferguson, Hongyan Shih, Nigel Goldenfeld and faculty members of the Center for the Physics of Living Cells for useful discussions. Alvaro Hernandez at the Roy J Carver Biotechnology Center at the University of Illinois at Urbana-Champaign performed the HiSeq sequencing and Elizabeth Ujhelyi provided assistance with MiSeq sequencing.

## Additional information

### Funding

| Funder | Grant reference number | Author |
|---|---|---|
| National Science Foundation | PHY 0822613 | David T Fraebel<br>Harry Mickalide<br>Diane Schnitkey<br>Jason Merritt<br>Thomas E Kuhlman<br>Seppe Kuehn |
| National Science Foundation | PHY 1430124 | David T Fraebel<br>Harry Mickalide<br>Diane Schnitkey<br>Jason Merritt<br>Thomas E Kuhlman<br>Seppe Kuehn |

The funders had no role in study design, data collection and interpretation, or the decision to submit the work for publication.

### Author contributions

DTF, Formal analysis, Validation, Investigation, Writing—original draft, Writing—review and editing; HM, Software, Formal analysis, Validation, Investigation, Writing—review and editing; DS, Investigation, Writing—review and editing; JM, Software, Writing—review and editing; TEK, Supervision; SK, Conceptualization, Data curation, Software, Formal analysis, Supervision, Funding acquisition, Investigation, Visualization, Methodology, Writing—original draft, Project administration, Writing—review and editing

### Author ORCIDs

David T Fraebel, http://orcid.org/0000-0002-8668-3476
Seppe Kuehn, http://orcid.org/0000-0002-4130-6845

## Additional files

### Major datasets

The following dataset was generated:

| Author(s) | Year | Dataset title | Dataset URL | Database, license, and accessibility information |
|---|---|---|---|---|
| Fraebel DT, Kuehn S | 2016 | Sequencing data for motility selection experiments | https://doi.org/10.13012/B2IDB-3958294_V1 | Publicly available at Illinois Data Bank (dataset ID: IDB-3958294) |

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

## Appendix 1

## Measurements of evaporative losses in agar plates

As discussed in Materials and methods, all plates for a single selection experiment were prepared from a single autoclaved bottle of medium and agar (0.3 % w/v). Plates were poured, allowed to cool, parafilmed and stored at 4°C. To initiate a new round of selection, plates were allowed to equilibrate at 30°C for 12 hr (rich medium) or 24 hr (minimal medium) in the environmental chamber prior to inoculation. We performed a series of control experiments by weighing plates to estimate the magnitude of the evaporative water loss due to storage, thermalization and the colony migration assay. For plates stored at 4°C for 9 days and then used in a round of selection we found the total evaporative losses would increase the agar concentration by 5% (e.g. from 0.3 % w/v to 0.315 % w/v). Using the data shown in *Figure 4* of *Croze et al. (2011)*, this corresponds to a change in migration rate of 0.06 cm h$^{-1}$. We take this as an upper bound on the uncertainty in migration rate in the rich medium condition due to uncontrolled variation in the agar concentration due to evaporative losses.

In minimal medium evaporative losses due to storage are similar to rich mediu but losses during colony expansion are diminished due to saturating humidity over the 48 hr expansion (Materials and methods). We therefore expect the fractional change in agar concentration to be similar between the two experiments. We measured migration rates for the founder strain in soft agar at two concentrations (0.2% and 0.3%) to estimate the change in migration rate as a function of agar concentration. From this we estimate that the error in migration rate due to evaporative losses in the plates during storage and migration are approximately 0.005 cm h$^{-1}$.

## Additional selection experiments

We performed selection on the motile but non-chemotactic mutant Δ*cheA-Z* in the same genetic background used for all other experiments (MG1655-motile). In rich medium, we observed migration an order of magnitude slower than the wild-type and only a small increase in $s$ over 10 rounds of selection (*Figure 1—figure supplement 1*). In this experiment each round of selection lasted 24 hr to permit this strain to form colonies large enough for reliable sampling. In minimal medium the non-chemotactic mutant formed no measurable front during 48 hr of incubation and selection, performed by sampling from the periphery of this colony, resulted in only a very small increase in migration rate in one replicate after seven rounds of selection. For the minimal medium experiment, antibiotics were used to limit the possibility of contamination and the Δ*cheA-Z* deletion was confirmed by PCR.

We performed selection in rich medium where populations were sampled every 12 hr from a point halfway between the center of the colony and the outer edge; results are shown in *Figure 1—figure supplement 3*. When sampling at this location we observed slower adaptation and a reduction in the maximum rate of expansion compared to populations selected by sampling at the migrating front.

Since previous work has shown that non-genetic diversity can be important in chemotaxis and front migration (*Frankel et al., 2014*; *Løvdok et al., 2007*), we asked whether long-term growth in liquid culture resulted in loss of the fast migration phenotype. We inoculated a strain isolated after 15 rounds of selection in rich medium (*Figure 1c*, main text, replicate 1) into a custom turbidostat that maintained a population of ∼10$^9$ cells under well mixed and constant temperature conditions. We inoculated soft agar plates from this continuous culture at regular intervals over approximately 140 generations of growth in liquid culture. We observed no decrease in the rate of migration due to prolonged growth in liquid culture

(*Figure 1—figure supplement 5*), suggesting that non-genetic variation likely does not play a large role in the adaptation we observed. Whole genome sequencing revealed that all mutations observed in the round 15 strain were retained after 40 generations of liquid culture (*Appendix 1—table 1*).

**Appendix 1—table 1.** Mutations present after 40 generations of liquid culture growth for rich medium replicate 1 round 15 strain.

**Rich medium, round 15 rep. 1, after 40 generations in liquid culture.**

| # gen. (coverage) | 40, (187×) |
|---|---|
| Mutation (loc., mut., frac., cov.) | 457978, *clpX* E185*, 100%, 254 |
| | 523086, IG Δ1 bp, 100%, 256 |
| | 990379, IG A→C, 100%, 165 |
| | 663115, Δ1 bp *dacA* FS, 100%, 161 |

## Measurements of growth rates

Growth rate measurements were performed using custom-built optical density measurement device (*Merritt and Kuehn, 2016*). Briefly, this device used an infrared LED and a photodetector to measure the transmitted light passing through a culture vial. The LED and photodetector were embedded in an aluminum block that was temperature controlled by a Peltier element and PID feedback software. Strains were inoculated from overnight culture into 20 ml vials of the appropriate medium stirred at 850 rpm and maintained at 30°C. The growth rate was measured by linear regression of $log(OD(t))$ over a 150 to 200 min window where the change in OD is determined to be exponential by inspecting the residuals. We checked that the conclusions in *Figure 3*, *Figure 3—figure supplement 3* and *Figure 4—figure supplement 2* did not depend qualitatively on the time interval used in fitting the optical density curves.

## Numerical simulations of reaction-diffusion model

Under the assumptions of vertical uniformity in the plate and azimuthal symmetry, the numerical integration of *Equations 1 and 2* in the main text was coded in C++ as a one dimensional lattice representing a horizontal line projecting from the center of the plate to the edge. Each lattice site had both a food/attractant density ($c(\mathbf{r}, t)$, initially uniform) and a bacterial surface density ($\rho(\mathbf{r}, t)$, with an initial inoculum corresponding to $1.4 \times 10^8$ cells ml$^{-1}$ at the center). A lattice spacing of 0.15 mm was used with a step time of 0.0625 min; every step the entire system was updated according to the model (in cylindrical coordinates) using standard nearest-neighbor finite difference equations for the first and second derivatives on a lattice. To prevent seeding the far end of the plate with bacteria in nonphysical time, densities greater than 100 cells ml$^{-1}$ were required to seed a lattice site as the bacteria propagated outward. Changing this threshold did not alter the results. The front position was determined by finding the first local maximum in $\rho$ from the edge of the plate. Front velocities were determined via linear fit on front position with time. Examples of simulation outputs are shown in *Figure 2—figure supplement 1*. Parameters for our simulation in both rich and minimal medium were either measured or taken from the literature and values are given in *Table 1* of the main text.

# Relationship between single-cell behavior and front migration

The relationship between single-cell swimming parameters ($|v_r|$, $\tau_r$ and $\tau_t$) and population transport parameters ($k_0$ and $D_b$) has been described in detail elsewhere (**Rivero et al., 1989**; **Croze et al., 2011**; **Celani and Vergassola, 2010**). Here we summarize the results of these calculations and give details for the estimates given in the main text. **Rivero et al. (1989)** considered chemotaxis in one spatial dimension by considering the dynamics of two populations of cells: those moving left and those moving right at constant speed $|v_r|$. They neglect variation in $|v_r|$ across the population and time; they also treat tumbles as instantaneous. They define the probability that a cell swimming to right tumbles and begins swimming to the left as $\alpha^+$ and the converse as $\alpha^-$. Under these assumptions they show that

$$D_b = \frac{2v_r^2}{\alpha^+ + \alpha^-} \tag{5}$$

$$k_0 = v_r \frac{\alpha^- - \alpha^+}{\alpha^+ + \alpha^-}. \tag{6}$$

Note that the tumble frequency is $\alpha_0 = \alpha^+ + \alpha^-$. As discussed in the main text, Croze *et al.* use this as a starting point for deriving a relationship between the transport parameters $D_b$ and $k_0$ and the behavioral parameters $|v_r|$ and $\tau_r$. For completeness we give the main results of their derivation here; for further details see Appendix A of **Croze et al., 2011**. First, *E. coli* modifies its tumble frequency in response to environment according to

$$\alpha(t) = \alpha_0[1 - \int_{-\infty}^{t} dt' K(t - t') f_{k_0}(x(t'))] \tag{7}$$

where $\alpha_0$ is the unstimulated tumble frequency, $x$ is a spatial coordinate, the integral contains the response function ($K$) (**Segall et al., 1986**) and $f_{k_0} = c(x)/(c(x) + K_D))$ describes the binding of an attractant at concentration $c(x)$ to the relevant receptor. Experimentally, it has been shown that $\int_{-\infty}^{\infty} dt K(t) = 0$.(**Segall et al., 1986**) We proceed by assuming that the effective tumble frequency due to collisions with the agar can be written as $\alpha_e(t, C) = \alpha(t) + \alpha_A(C)$. The authors then compute an average run duration given $\alpha_e(t, C)$. We note that in this derivation, $f_{k_0}$ is expanded and truncated to first order. The result is

$$D_b(C) = \frac{v_r^2}{\alpha_0} \frac{1}{(1 + \alpha_A(C)/\alpha_0)} \tag{8}$$

$$k_0(C) = \frac{v_r^2}{\alpha_0} \frac{1}{(1 + \alpha_A(C)/\alpha_0)^2} \int_0^{\infty} dt K(t) e^{-(\alpha_0 + \alpha_A(C))t} \tag{9}$$

For $k_0$ the authors approximate the integral when $\alpha_A(C) \approx \alpha_0$ (the 'efficient limit') to $\int_0^{\infty} dt K(t) e^{-(\alpha_0 + \alpha_A(C))t} \approx 1 - \kappa \alpha_A(C)/\alpha_0$ with $\kappa = \int_0^{\infty} dt' \alpha_0 t' K'(t') e^{-\alpha_0 t'} / \int_0^{\infty} dt' K(t') e^{-\alpha_0 t'}$. Using a previously proposed parameterization of $K(t) = N_0 e^{-\alpha_0 t}(1 - A_0(\alpha_0 t + 1/2\alpha_0^2 t^2))$(**Clark and Grant, 2005**), we find that $\kappa = 0.1$ ($A_0 = 0.5$). Thus,

$$k_0(C) \approx \frac{v_r^2}{\alpha_0} \frac{(1 - \kappa \alpha_A(C)/\alpha_0)}{(1 + \alpha_A(C)/\alpha_0)^2} \tag{10}$$

The authors then postulate that

$$\alpha_A(C) = \alpha_0 e^{(C-C_1)/C_0} \tag{11}$$

and empirically determine the constants $C_1$ and $C_0$ by fitting the measured dependence of front migration rate on agar concentration. They compute $C_1 = 0.28\%$ and $C_0 = 0.035\%$. They show that the efficient limit described above captures the dependence of the rate of migration on agar concentration as well as changes in the shape of the front due to changes in agar concentration. Using **Equations 8, 10 and 11** for our conditions ($C = 0.3\%$) with previously measured values of $D_b$ and $k_0$ in liquid (**Ahmed and Stocker, 2008**), we estimate $D_b$ and $k_0$ in the presence of agar for both rich and minimal medium conditions

To generate the heat maps shown in **Figures 2,4** of the main text, we varied $|v_r|$ and $k_g$. Tumble frequency ($\alpha_0$) was assumed fixed for these simulations since changes in tumble frequency alone were found to drive only small changes in front migration rate (**Figure 2—figure supplement 3**). We therefore recomputed $D_b$ and $k_0$ for each value of $|v_r|$ and $k_g$ and performed a simulation of front migration.

To estimate how the evolution of run tumble statistics at the single-cell level (**Figure 3**, main text) in liquid changed $D_b$ and $k_0$, we assumed $K(t)$ and $\alpha_A$ were unchanged by selection. We recomputed **Equations 8 and 10** using the observed changes in $\alpha_0$ and $|v_r|$ (**Tables 3** and **4**, main text). We then simulated **Equations 1 and 2** from the main text using these values and the measured change in growth rates (**Figure 3g–h**, main text). We found that these changes predicted an increasing rate of migration with selection which was qualitatively correct (**Figure 4—figure supplement 1**). We note that our single-cell measurements were made in a uniform environment without spatial gradients in attractants and we therefore cannot determine whether or not $K$ has changed during selection.

## The effect of boundary interactions in microfluidic device on run-tumble statistics

When *E. coli* swims very close to a surface, interactions between the helical bundle and the surface can result in cells swimming in circles (**DiLuzio et al., 2005**; **Frymier et al., 1995**). However, wild type cells execute tumbles even in the presence of surfaces (**Frymier et al., 1995**) and previous methods for tracking single-cells similar to ours have found that cells exhibit typical run-tumble behaviors even in microfluidic devices with a floor to ceiling height as small as 4 µm (**Umehara et al., 2007**). Our chambers are 10 µm in depth and we typically observe run-tumble behavior similar to that shown in **Figure 3—figure supplement 1**. However, we did transiently observe cells 'circling' likely due to close proximity to the floor or ceiling of the chamber. To check that this circling behavior was not biasing our results, we devised an automated technique to detect circling. For each run event longer than 10 frames we compute the sign of $\omega(t)$ for each frame included in the run, which we denote $sign(\omega_{run}(t))$. For each run we compute $\xi = |\langle sign(\omega_{run}(t)) \rangle|$. $\xi$ is close to unity for cells that are circling and close to zero for cells that are not circling. By inspection of trajectories we determined that cells with $\xi > 0.17$ more than 65% of their entire trajectory could be regarded as circling. This classified approximately 15% of the data as circling due to boundary interactions. The data shown in **Figure 3** and all supplemental figures excludes these circling cells. However, we checked that the conclusions of our study, most importantly changes in run speed, were not qualitatively altered even when we included circling cells in our analysis.

## Comparison of rich medium round 15 strain with RP437

We tested whether or not the strain selected for fast migration in rich medium differed substantially from the RP437 strain most commonly used in chemotaxis studies. We measured the migration rate for RP437 to be $0.15 \pm 0.009$ cm h$^{-1}$ in rich medium, approximately a factor of two slower than MG1655-motile (founder strain) in identical conditions. We observed similar single-cell behavioral statistics between the two strains (*Figure 1—figure supplement 4*) so we attributed slower migration to the reduced growth rate of RP437 relative MG1655-motile ($1.1 \pm 0.02$ h$^{-1}$ and $1.24 \pm 0.02$ h$^{-1}$ respectively) measured in well mixed liquid culture.

## Experimental details of mutant reconstruction

To reconstruct point mutations in the chromosome of the founding strain, we followed a method described in *Kuhlman and Cox (2010)* and outlined in the Materials and methods section of the main text. Here we detail the experimental methods used in this reconstruction.

### Preparation and electroporation of electrocompetent cells

0.5 mL of an overnight culture was added to a flask containing 30 mL of LB with appropriate antibiotic(s) and inducer(s) and grown at 30°C with shaking until the OD600 reached 0.5 to 0.7. The flask was removed and the culture was cooled by swirling in an ice water slurry for five minutes then placed on ice for ten minutes. The culture was transferred to a pre-chilled centrifuge tube and pelleted by centrifugation (5 min, 5K RPM) in a refrigerated centrifuge chilled to 4°C. The supernatant was dumped and the cells were washed in 10 mL of ice cold 10% glycerol. Glycerol wash was repeated twice followed by a resuspension in 200 $\mu$L. The cells were immediately placed on ice and kept cold until electroporation. Typically, $\sim$100 $\mu$L of cells was mixed with $\sim$5 $\mu$L of DNA in a pre-chilled microcentrifuge tube before being transferred to a pre-chilled 0.1 cm gap electroporation cuvette (USA Scientific) and electroporated at 2 kV in an Electroporator 2510 (Eppendorf).

### Synthesis and integration of the landing pad

Custom primers were designed and ordered from Integrated DNA Technologies. These primers contain homologies to *tetA* flanking regions on template plasmid pTKLP-*tetA* as well as 50 bp homologies just upstream/downstream of the desired chromosomal mutation site. PCR using these primers generated the linear landing pad fragment for each desired mutation site. Purification was performed with AmpureXP magnetic beads followed by a DpnI digest and an additional AmpureXP cleanup. Fragment length was confirmed by 1% agarose gel. Electrocompetent founder+pTKRED cells were prepared from frozen stock, with 2 mM IPTG induction of the $\lambda$-Red enzymes on pTKRED. These cells were transformed with the landing pad fragment. After 4 hr outgrowth on the bench, half the culture was pelleted in a microcentrifuge (1 min, 14000 RPM). 410 $\mu$L of the supernatant was removed, cells were resuspended in the remaining media and plated on LB+tetracycline +spectinomycin plates. The remaining half of the culture was plated in the same way after an additional day of outgrowth on the bench. The plates were grown at 30°C and colonies typically took a few days to appear at this step. PCR and 1% agarose gel electrophoresis on resultant colonies was used to confirm successful landing pad integration at the desired site. The presence of a secondary band consistent with the wild-type revealed heterogeneity within transformant colonies. The landing pad strain was therefore purified by overnight growth (30°C, shaking) in LB+tetracycline+spectinomycin followed by serial dilution and plating.

## Integration of desired mutation

A 70 bp oligo containing each desired mutation was designed following the design considerations outlined in *Sawitzke et al., 2013* as closely as possible and ordered from Integrated DNA Technologies. Electrocompetent founder+pTKRED+LP cells were prepared from frozen stock, with 2 mM IPTG induction of the $\lambda$-Red enzymes as well as 0.4 % w/v L-Arabinose induction of Isce-I from pTKRED. These cells were electroporated with the oligo and 1 mL of LB was immediately added. The cells were then transferred to a flask containing 100 mL of RDM+0.5 % glycerol with inducers and spectinomycin. The flask was grown for 2 hr at 30°C with shaking before adding $NiCl_2$.

The appropriate $NiCl_2$ concentration was determined in a separate experiment wherein growth of founder+pTKRED as well as founder+pTKRED+LP was assayed in the supplemented RDM described above. At each day of the outgrowth until successful transformants were identified, a sample was diluted and plated on LB+spectinomycin. Colonies from these plates were screened for tetracycline resistance. A few tetracycline-susceptible colonies were checked for successful landing pad removal by colony PCR and 1% agarose gel electrophoresis. Finally the pTKRED plasmid was cured by growth at 42°C. Mutations were confirmed by Sanger sequencing.

## Modeling evolution of correlated traits

The model of evolving correlated traits is derived in detail elsewhere (*Lande, 1979*). We infer constraints on entries in the matrix $G$ by comparing the predicted ($\hat{\phi}_{pred}$) direction of phenotypic evolution to that which we observed ($\hat{\phi}_{obs}$). We determined the observed direction of phenotypic evolution by linear regression of the data shown in *Figure 4a–b* of the main text. We then compute the dot product $\hat{\phi}_{pred} \cdot \hat{\phi}_{obs}$ over a range of values of $\sigma_{\bar{k}_g}$, $\sigma_{|\bar{v}_r|}$ and $\rho$ (*Figure 6—figure supplement 2*).

We note that the migration rate in minimal media as a function of $|v_r|$ and $k_g$ contains a strong nonlinearity around a growth rate of 0.2 h$^{-1}$. This transition occurs between regimes where bacterial transport is dominated by growth and diffusion (founder) and chemotaxis (evolved) (*Croze et al., 2011*). The characteristic timescale for the migration process is set by the growth rate $\tau \sim 1/k_g$ and the length scale by the distance a cell diffuses over its lifetime $l \sim \sqrt{D_b/k_g}$. For the founding strain in minimal medium, $\tau \sim 10$ hr while $l \sim 0.5$ cm. In this case $\nabla c$ remains small and transport is dominated by diffusion and subsequent growth. As growth rates increase during selection and $D_b$ decreases only modestly (see *Table 4*, main text) and $\tau \sim 3$ hr and $l \sim 0.15$ cm. In this case chemotactic transport becomes substantial due to gradients that form over lengthscale $l$ and we observe this transition to migrating fronts around $k_g \sim 0.2$ h$^{-1}$. We note that this transition predicted by our model is also observed experimentally (*Figure 1e*, main text).

However, as a result of this non-linearity our estimate of $\vec{\beta}$ in minimal medium relies on a poor linear fit (*Figure 6—figure supplement 1*). To asses whether or not this poor approximation might alter our conclusions, we performed stochastic simulations of an evolving population that did not require us to make a linear approximation to infer $\vec{\beta}$. To accomplish this, we generated a population of 1000 individuals whose phenotype was drawn from the multivariate normal distribution $\mathcal{N}(\vec{\phi}^f, G)$ where $\vec{\phi}^f$ is the mean phenotype of the founding population and $G$ is the genetic covariance matrix discussed in the main text. Using the predicted migration rate as a function of $|v_r|$ and $k_g$ as a fitness landscape (*Figure 2b*, main text) we then select that fastest 1% of the population. From this selected population we compute a new $\vec{\phi}^1$ and generate a second population from the distribution $\mathcal{N}(\vec{\phi}^1, G)$. The process is repeated iteratively. The results of these simulations are shown in *Figure 6—figure supplement 3*. We find that our qualitative conclusions hold. Large

negative values of the correlation coefficient ($\rho < -0.9$) and $\sigma_{\tilde{k}_g} > \sigma_{|\tilde{v}_r|}$ result in evolution in the same direction we observe experimentally. We note that in these simulations populations with finite $\sigma_{|\tilde{v}_r|}$ and $\rho > -1$ are able to evolve both higher run speeds and growth rate.

## Estimated change in drag due to change in growth rate

For a bacterium swimming at constant speed $u$ (at low Reynolds number) the propulsion force provided by the flagella ($F_{flag}$) equals the drag force from the fluid ($F_D$). Thus, we can write:

$$u = \frac{F_{flag}}{F_D/u} \tag{12}$$

The ratio of swimming speeds in a given medium for evolved and founding strains is therefore:

$$\frac{u^{ev}}{u^f} = \frac{F_{flag}^{ev}\left(F_D^{ev}/u^{ev}\right)^{-1}}{F_{flag}^{f}\left(F_D^{f}/u^{f}\right)^{-1}} \tag{13}$$

If we assume the flagellar force is unchanged with selection, then we have:

$$\frac{u^{ev}}{u^f} = \frac{F_D^{f}/u^f}{F_D^{ev}/u^{ev}} \tag{14}$$

The drag force on an ellipsoid moving along its symmetry axis at speed $u$ in a fluid with viscosity $\mu$ is given by equation (2.3) in *Cox, 1970*:

$$F_D = 16\pi\mu au\left[\frac{-2\chi}{\chi^2-1} + \frac{2\chi^2-1}{(\chi^2-1)^{\frac{3}{2}}} + \ln\frac{\chi+\sqrt{\chi^2-1}}{\chi-\sqrt{\chi^2-1}}\right]^{-1} \tag{15}$$

Where $\chi$ is the ratio of the major axis (half-length) $b$ to the minor axis (half-width) $a$.

$$\chi = \frac{b}{a} = \frac{l/2}{w/2} = \frac{l}{w} \tag{16}$$

It can be shown that (*Equation 11*) is equivalent to:

$$F_D = 6\pi\mu au K' \tag{17}$$

where

$$K' = \frac{\frac{4}{3}(\chi^2-1)}{\frac{2\chi^2-1}{\sqrt{\chi^2-1}}\ln\left[\chi+\sqrt{\chi^2-1}\right]-\chi} \tag{18}$$

Using the above, we have:

$$\frac{u^{ev}}{u^f} = \frac{w_f K'_f}{w_{ev} K'_{ev}} \tag{19}$$

Figure S1A of *Taheri-Araghi et al., 2015* gives the average length and width of an *E. coli* as a function of its growth rate:

$$l = 2.08 * 2^{0.41 * \frac{divisions}{hour}} \mu m = 2.08 * 2^{0.41 * \frac{r}{\ln 2}} \mu m \tag{20}$$

$$w = 0.41 * 2^{0.36 * \frac{divisions}{hour}} \mu m = 0.41 * 2^{0.36 * \frac{r}{\ln 2}} \mu m \tag{21}$$

Using this expression for the width, we have:

$$\frac{u^{ev}}{u^f} = \frac{K'_f}{K'_{ev}} * 2^{0.519(r^f - r^{ev})} \tag{22}$$

Where $r^f$ and $r^{ev}$ are the founder and evolved growth rates respectively. From our growth rate experiments, we have:

$r^f_{LB} = 1.24 \text{ h}^{-1}$; $r^{ev}_{LB} 0.08 \text{ h}^{-1}$

$r^{ev}_{LB} = 1.09 \text{ h}^{-1}$ ; $r^{ev}_{gal} = 0.40 \text{ h}^{-1}$

Using these values, we can calculate $\chi$ (and therefore K') from *Equations 16 and 17* and plug these into (*Equation 18*). Doing this, we find that:

$$\frac{u^{ev}_{LB}}{u^{founder}_{LB}} = 1.059$$

$$\frac{u^{ev}_{gal}}{u^{founder}_{gal}} = 0.884$$

We see that the change in drag due to the change in cell size that we calculate using (*Equations 16 and 17*) and our growth rate data would only account for a 6% swimming speed increase in LB (rich medium) and a 12% swimming speed decrease in galactose (minimal medium). We note that the growth rates of our strains in rich medium (LB) lie within the range of growth rates measured by Taheri-Araghi *et. al*, however the growth rates in galactose minimal medium are significantly slower. Finally, since we have not measured cell size in our evolved strains we cannot definitively rule changes in size as a mechanism for the trade-off observed here.

