## [Decision Letter]

[Editors’ note: a previous version of this study was rejected after peer review, but the authors submitted for reconsideration. The first decision letter after peer review is shown below.]

Thank you for submitting your work entitled "Evolution of bacterial motility through a porous environment" for consideration by *eLife*. Your article has been favorably evaluated by Diethard Tautz (Senior Editor) and three reviewers, one of whom is a member of our Board of Reviewing Editors.

The reviewers have discussed the reviews with one another and the Reviewing editor has drafted this decision. Given the additional extensive revisions that the reviewers argue must be accomplished, we feel that we must reject the current submission. Under the circumstances, we would understand if you now chose to submit this work elsewhere. However, if you feel you are able to address the concerns expressed by the reviewers, you may wish to have us consider this work again in which case we would be happy to consult the same reviewers on such a new submission.

Summary:

"Evolution of bacterial motility through a porous environment" suggests that bacteria can evolve faster motility in a porous environment by executing shorter runs (avoiding collision with agar), swimming faster, and reducing cell-to-cell variability (the evolved populations had a lower growth rate than the ancestor population). The experiment was done through selecting for edge (faster-migrating) cells migrating through low-percentage agar. Given that it has been known for a long time that shorter run durations are advantageous in a porous environment (Migration of bacteria in semisolid agar. A J Wolfe and H C Berg.Proc Natl Acad Sci USA, 1989 vol. 86 (18) pp. 6973-6977), the conclusions are not surprising. Thus, we would like you to put less emphasis on this less novel aspect and acknowledge so (e.g. explicitly stating that the result is as expected). Reduction in cell-to-cell variability is potentially interesting, but we are not sure whether this is a direct consequence of faster motility. Moreover, the significance of this result seems to depend on the dataset that the observations came from (see reviewer 3 comments below).

While the authors identified some candidate mutations to explain the observed changes in behavior, they did not test any of them in an ancestral background. Thus, we don't know which mutations were necessary for the altered phenotypes. It would be very interesting to know, for example, whether one mutation was sufficient to reduce variance in the population, while another was sufficient to reduce run duration, etc. This kind of analysis would provide an obvious step forward because it would provide an understanding of how motility can be rapidly altered by selective pressures at the molecular/genetic level.

*Reviewer #1:*

1) Does the evolution of faster motility require a porous environment? If you propagate *E. coli* in a well-mixed environment (e.g. in a chemostat selecting for faster growth), will you see faster motility as a byproduct of faster growth (mutations can be pleiotropic)? More critically, if you inject *E. coli* in the middle of a static broth environment (no agar) and select for cells at the peripheral for >50 generations, do you expect to see faster motility? If you see faster motility, will cells not show shorter runs?

2) The authors need to do a better job characterizing growth phenotypes, especially since *E. coli* experience a range of growth conditions in a spatially-structured environment. I suspect that the observation that Gen 15 evolved cells show slower growth rate than ancestral cells in abundant nutrients is linked to the possibility that they may grow better than ancestral cells in limited nutrients. This kind of fitness tradeoff has been seen in many systems. You can do a chemostat competition experiment to test this. Thus, evolved cells may "beat" ancestors in two ways: they can arrive at abundant LB first, so their slower growth in abundant LB will not make much difference; and they can also grow better than the ancestor when stuck at the inner region. This may help explain how quickly they come to fixation.

*Reviewer #2:*

1) I particularly like the approach taken by the authors to measure single-cell swimming behavior, but wonder how the authors keep track of cells if they move out of focus. Do the authors focus on cells near glass surface? In that case cell-surface interaction will cause cells to swim in circles and suppress tumbling, which may obscure the results.

2) The authors identify tumbles based on angular velocity, "Tumbles were initiated whenever ω(t) > 6 rad s^−1^ and continued until ω(t) < 3.9 rad s^−1^. These two thresholds on ω were determined by eye and resulted in average run and tumble durations in accordance with previously published values." This criterion is different from that used in some other publications, such as Berg and Brown, 1972, where tumbles were identified by both abrupt change of velocity direction and reduction of speed (due to complete or partial loss of propulsive force when flagellar bundle disassembles). The authors obtained a mean run time of 0.42 ± 0.005 s for WT cells, almost half as much as that obtained in Berg and Brown, 1972 (0.86 s) for AW405 cells swimming in bulk fluids. I suppose this two-fold difference in run time could not be attributed to strain difference, but most likely due to the difference in definition of tumbles.

3) RP437 and AW405 strains were selected for spreading and chemotaxis capability in swimming agar plates from *E. coli* K-12 strain. Do RP437 or AW405 behave similarly to the evolved strains in the paper (i.e. executing runs with faster speeds but with reduced duration compared to MG1655)?

*Reviewer #3:*

1) In general, I enjoyed the writing style and the attention to detail in the manuscript. In particular, the experiments were solid and the controls were thorough, especially for Figure 2. However, I have trouble understanding why the authors did not propagate all lines to 15 rounds since it only involves doing 5 more plates for each line. This is especially important since a lot of subsequent analysis is carried out using the "evolved strain" that harvested from that single line of evolution after 15 rounds. Having more lines would also indicate whether or not the reduction in diversity is a general feature.

2) The modeling provides nice qualitative support to the findings about the relationship between front speed and geometry and the cells parameters but maybe takes too much place in the paper since many of the parameters were poorly constrained or taken from experiments done on other strains in different conditions. The authors are up front about this, and do a good job explaining the model, but it seems that the model is just used to say that the observed changes are probably not due to changes in growth (which they show experimentally anyway), and that there are probably other factors not accounted for in the model that are responsible for improved performance. Space could be used instead for addressing concerns mentioned above.

3) In Figure 1.the profile of intensity appears to get gradually lower from the center, and then abruptly lower to form the outer ring. I don't see how this can be compared to the simulated profile, which appears to gradually increase in intensity from the center, before forming an abruptly higher intensity ring. Is this why the profiles in Figure 3—figure – supplement 1A and D look so different from one another? Can you account for these differences? Is it due to uneven illumination in the experiment? Did the authors use an inverse scaling for the Figure 4 (dark is more cells rather than light is more cells)?

4) In the Results, the authors say that "the increase in s was reproducible across independent selection experiments", but it is hard to tell without showing the trajectory of each experiment. Figure 2 shows that on average, s increased. This is especially interesting because in Figure 3, the round where one experiment is missing has a much smaller error bar than the other rounds. This suggests that one of the experiments was quite different from the others. Was the missing observation from the experimental replicate that had a different mutational trajectory?

5) Since tracks were cut when cells interacted with the boundary, how where the runs that were not terminated on both ends by tumbles treated (or tumbles that were not terminated on both ends by runs)? These runs (tumbles) would be of undefined length. Was the data analyzed without these undefined events?

6) Variance in the S dataset is in general greater than it is for the L dataset. The authors explain that this is due to sampling fewer run-tumble events for each individual in the S dataset. From the description, S individuals were observed for ~5 min, while L individuals were observed for ~10 min. However, there were at least 6x more individuals in S. Could the larger population size account for the increased variance? In general to assess cell-to-cell variability one should measure many cells rather than a few cells for a very long time. On a related note, the reduction in variance seems to be only significant (according to Figure 6 for one of the two datasets in two of the three measured parameters for which a reduction in variance is claimed (S for run speed, L for run time). Ideally, the significance of the reduction should not depend on the dataset.

[Editors’ note: what now follows is the decision letter after the authors submitted for further consideration.]

Thank you for submitting your article "Environment determines evolutionary trajectory in a constrained phenotypic space" for consideration by *eLife*. Your article has been favorably evaluated by Diethard Tautz (Senior Editor) and three reviewers, one of whom, Wenying Shou (Reviewer #1), is a member of our Board of Reviewing Editors. The following individuals involved in review of your submission have agreed to reveal their identity: Yilin Wu (Reviewer #2), and Thierry Emonet and Adam Waite (Reviewer #3).

The reviewers have discussed the reviews with one another and the Reviewing Editor has drafted this decision to help you prepare a revised submission.

Summary:

Fraebel et al. selected *E. coli* for faster migration through porous environment (low% agarose) in rich or minimal medium. Their qualitative mathematical model predicted that faster growth rate and faster run speed during chemotaxis are the most important elements in achieving faster migration through agar. They observed an evolutionary tradeoff between run speed and growth rate. In rich medium, mutations leading to improved run speed at the cost of slower growth are selected, whereas in minimal medium, mutations leading to improved growth rate at the cost of slower run speed are selected. Single mutations display these tradeoffs, suggesting antagonistic pleiotropy. Their mathematical model suggests that in rich versus minimal medium, the relative variances of the two phenotypes (growth rate and run speed) differ, which causes different evolutionary trajectories.

We all feel that the work is solid and interesting. However, we all feel that your narratives can be modified to achieve a greater level of clarity, especially regarding model-experiment comparison. We are all aware of the difficulty in modeling biological systems (especially dynamics of biological systems). Thus, a lack of great fit is not too surprising, but we do expect well thought-through explanations or even speculations on what your model explains or fails to explain.

Thus, we invite you to revise your writing in accordance with our suggestions.

*Reviewer #1:*

Although tradeoff between run speed and growth rate has been observed before, I do like the contrast between different outcomes in different environments. Overall, I find the paper solid and interesting. However, their narratives can be modified to achieve a greater level of clarity. I list a few examples below.

1) Authors claimed that in both rich and minimal media, their five lines evolved similarly. I am not sure that I agree with their assessment, although without error bars in graphs (Figure 1), it is tough to tell one way or another. I suggest authors add error bars (or an estimation of errors). Authors could also simply write something like "migration rates increased in all lines, though the extent of improvement differed along lines".

2) In some lines, despite selection, migration rate seemed to decline at later stages. Is that caused by genetic drift, as described by for example "Genetic drift at expanding frontiers promotes gene segregation" by Hallatschek et al.? This is a striking feature of the graph, and in my opinion, should be touched upon even if briefly.

3) You often claimed that your model qualitatively captured experiments (e.g. Figure 2—figure supplement 1). I am not sure which qualities you are referring to, especially with respect to rich medium experiments. The agreement is pretty poor in my opinion, unless I am missing something.

*Reviewer #2:*

This manuscript aims to address how constraints on phenotypic variation may limit the capacity of organisms to adapt to the multiple selection pressures, using *Escherichia coli* colony expansion in a porous environment as a model system. The authors found that a trade-off between swimming speed and growth rate that depends on the environment (rich medium versus minimal medium, in this case). They further showed that the trade-off is mediated by antagonistic pleiotropy through mutations that affect negative regulation. The paper is well-written and the results are clearly presented.

Evolutionary dynamics under multiple selection pressures have been investigated by a number of studies, some of which were cited in the manuscript. The main novelty of this paper, as the authors pointed out, is in that the selection process here involves multiple stresses simultaneously. In this regard, I hope the following point could be clarified: By selecting populations at migration front, one clearly imposes selection pressure on migration rate; but it is less clear to me whether a selection pressure on growth rate exists. Having a higher or lower growth rate does not necessarily guarantee whether a subpopulation can reach the edge, or for the specific population at colony edge, growth rate and migration rate could be uncoupled. *Escherichia coli* colony spreading through soft agar indeed depends on both motility and growth, but this is at the level of entire colony. Another way to impose selection pressures simultaneously on growth rate and migration rate would be: (a) grow many plates in parallel; (b) select the plate with largest colony size; (c) pool all cells on that plate for next round of propagation and selection. (Reviewing editor's comments: you can discuss that in Discussions).

*Reviewer #3:*

The authors have done a good job addressing my concerns from their initial submission. They have made an initially strong and interesting paper even stronger and more interesting. I am especially impressed by the thoroughness of their controls.

In general I find the comparison between model and data to be difficult to follow throughout the paper. I found myself repeatedly lost jumping between main figures and sup figures. The end result is that I had the impression that the model was vaguely useful. I think authors could do a better job at clearly explaining what the model does and does not explain.

---

## [Author Response]

[Editors’ note: the author responses to the first round of peer review follow.]

*Summary:*

*"Evolution of bacterial motility through a porous environment" suggests that bacteria can evolve faster motility in a porous environment by executing shorter runs (avoiding collision with agar), swimming faster, and reducing cell-to-cell variability (the evolved populations had a lower growth rate than the ancestor population). The experiment was done through selecting for edge (faster-migrating) cells migrating through low-percentage agar. Given that it has been known for a long time that shorter run durations are advantageous in a porous environment (Migration of bacteria in semisolid agar. A J Wolfe and H C Berg.Proc Natl Acad Sci USA, 1989 vol. 86 (18) pp. 6973-6977), the conclusions are not surprising. Thus, we would like you to put less emphasis on this less novel aspect and acknowledge so (e.g. explicitly stating that the result is as expected).*

We have included a more detailed discussion of the Wolfe and Berg (1989) results as they pertain to our study. We have stated explicitly that reductions in run duration are known to be associated with increased rates of migration through porous environments. We have shown through simulation that the expected changes in migration rate from changes in tumble frequency alone are smaller than the increases we observe experimentally (Figure 2—figure supplement 2).

*Reduction in cell-to-cell variability is potentially interesting, but we are not sure whether this is a direct consequence of faster motility. Moreover, the significance of this result seems to depend on the dataset that the observations came from (see reviewer 3 comments below).*

Again, we have chosen to omit this result from our current study. However, Figure 7 shows that individuality does in fact decline during selection in rich medium. To determine this, we performed single cell tracking on ~450 individual cells.

Author response image 1.Evolution of individuality in rich medium: Individuality for founding strain as well as strains isolated after 5, 10 and 15 rounds of selection was computed as the standard deviation across individuals for four behavioral parameters: τr, στr, τt and |vr |.Standard deviations in these parameters were computed for the 140 (founder), 79 (round 5), 97 (round 10) and 96 (round 15) individuals. Error bars are 95% confidence intervals from bootstrapping. The decline in standard deviation with selection is statistically significant for all four parameters (p < 0.005 (a-d), permutation test). (**e**) Measures individuality by a method that does not require the classification of runs and tumbles (Jordan et al. 2013). Jensen-Shannon divergence is computed between histograms of swimming speed and angular velocity for all pairs on individuals from a given strain. Increasing Jensen- Shannon divergence denotes higher levels of individuality. The decline in individuality between founder and round 5 is significant as well as between round 5 and round 10 and round 10 and round 15 (p < 0.001 for all comparisons, rank sum test).**DOI:**
http://dx.doi.org/10.7554/eLife.24669.039

*While the authors identified some candidate mutations to explain the observed changes in behavior, they did not test any of them in an ancestral background. Thus, we don't know which mutations were necessary for the altered phenotypes. It would be very interesting to know, for example, whether one mutation was sufficient to reduce variance in the population, while another was sufficient to reduce run duration, etc. This kind of analysis would provide an obvious step forward because it would provide an understanding of how motility can be rapidly altered by selective pressures at the molecular/genetic level.*

We have studied the phenotypes of mutations in the ancestral background for both rich medium and minimal medium conditions. Using scarless recombineering, we engineered point mutations in *clpX* and *galS* as well as an intergenic single base pair deletion. We have also studied the phenotypes of knockout mutants for *clpX* and *galS.* With respect to the speculation that different mutations might be responsible for changes in run speed and reduction in variability: we observe an increase in run speed for the *clpXE185** mutant concurrently with a decline in individuality in rich medium (data not shown for individuality). We concluded that these traits do not appear to be under independent genetic control.

*Reviewer #1:*

*1) Does the evolution of faster motility require a porous environment? If you propagate E. coli in a well-mixed environment (e.g. in a chemostat selecting for faster growth), will you see faster motility as a byproduct of faster growth (mutations can be pleiotropic)?*

To test this possibility, we inoculated the founding strain in a turbidostat and continuously cultured it for 200 generations in rich medium while periodically testing the front migration rate (Figure 1—figure supplement 2). We find that long-term growth in well-mixed liquid does not result in a large increase in front migration rate (~3% vs. ~100% for selection in low-viscosity agar). We performed an analagous experiment in minimal medium, where we continuously cultured cells for 100 generations (22 days) in minimal medium. In this case we did observe a substantial increase in the rate of migration in low-viscosity agar, but the increase was smaller than what we observe due to selection in plate (Figure 1—figure supplement 2).

*More critically, if you inject E. coli in the middle of a static broth environment (no agar) and select for cells at the peripheral for >50 generations, do you expect to see faster motility? If you see faster motility, will cells not show shorter runs?*

Testing the expansion of populations in pure liquid is very challenging since in the presence of even very small thermal gradients (which are exceedingly hard to suppress on centimeter length scales), transport will be dominated by convective flow. Therefore, we have not performed this experiment. We regard the question of how environmental structure (gel concentration) alters evolution to be an interesting question which is outside the scope of the current study.

*2) The authors need to do a better job characterizing growth phenotypes, especially since E. coli experience a range of growth conditions in a spatially-structured environment. I suspect that the observation that Gen 15 evolved cells show slower growth rate than ancestral cells in abundant nutrients is linked to the possibility that they may grow better than ancestral cells in limited nutrients. This kind of fitness tradeoff has been seen in many systems. You can do a chemostat competition experiment to test this. Thus, evolved cells may "beat" ancestors in two ways: they can arrive at abundant LB first, so their slower growth in abundant LB will not make much difference; and they can also grow better than the ancestor when stuck at the inner region. This may help explain how quickly they come to fixation.*

We have performed additional growth rate measurements of the founding strain and strains isolated from 5, 10 and 15 rounds of selection. To test the possibility that the evolved strains grows more quickly in nutrient limited conditions, we made growth rate measurements in diluted LB (10-fold and 100-fold dilutions, Figure 8). We find that the founder strain has a higher growth rate even under nutrient limited conditions. While we recognize that this is not a direct competition experiment, the growth rate difference is very large (30-50%).

Author response image 2.Growth rates of rich medium evolved strains under nutrient limitation.(left) is data reproduced from Figure 3 of the main text. (center) growth rates for founder (1) strain and strains after 5, 10 and 15 rounds of selection in 10-fold diluted LB medium. (right) identical measurements in 100-fold diluted LB medium. Declines in growth rate due to selection in 10- and 100-fold dilutions are statistically significant by linear regression (coefficient (confidence interval)):−0.019 [1/h* round] (−0.022, −0.015); −0.019 [1/h*round] (−0.031, −0.0005).**DOI:**
http://dx.doi.org/10.7554/eLife.24669.040

*Reviewer #2:*

*1) I particularly like the approach taken by the authors to measure single-cell swimming behavior, but wonder how the authors keep track of cells if they move out of focus. Do the authors focus on cells near glass surface? In that case cell-surface interaction will cause cells to swim in circles and suppress tumbling, which may obscure the results.*

We focus in between the PDMS and glass surface and neglect motion of the cells into and out of the focal plane – tracking behavior in 2-dimensions. With respect to circling due to interactions with the floor and ceiling of the chamber: we note that several previous studies have used a similar geometry to our and observed wild-type run-tumble statistics (e.g. Umehara et al.Biophysical Journal, 2008). However, we do observe transient circling in some cell isolated in our chambers. To check that this was not altering our results, we devised a quantitative method to detect circling in our single-cell trajectories (described in Supplementary file 1). When we remove individuals that exhibit some circling by this metric, we find that our conclusions are not altered.

*2) The authors identify tumbles based on angular velocity, "Tumbles were initiated whenever ω(t) > 6 rad s^−1^ and continued until ω(t) < 3.9 rad s^−1^. These two thresholds on ω were determined by eye and resulted in average run and tumble durations in accordance with previously published values." This criterion is different from that used in some other publications, such as Berg and Brown, 1972, where tumbles were identified by both abrupt change of velocity direction and reduction of speed (due to complete or partial loss of propulsive force when flagellar bundle disassembles). The authors obtained a mean run time of 0.42 ± 0.005 s for WT cells, almost half as much as that obtained in Berg and Brown, 1972 (0.86 s) for AW405 cells swimming in bulk fluids. I suppose this two-fold difference in run time could not be attributed to strain difference, but most likely due to the difference in definition of tumbles.*

To address this, we have implemented a different run-tumble detector based on a recent paper (Taute, Gude, Tans & Shimizu, Nat. Comm. 2015) which is based on a previous technique due to Berg and Brown (1972). The details of our implementation of this method are discussed in the Methods of the main text. We find this approach returns longer average run durations (0.5 – 0.7s) for our data. This detector is especially appropriate for our study because it readily accommodates cells with variable run speeds. We thank the reviewer for pointing this out.

*3) RP437 and AW405 strains were selected for spreading and chemotaxis capability in swimming agar plates from E. coli K-12 strain. Do RP437 or AW405 behave similarly to the evolved strains in the paper (i.e. executing runs with faster speeds but with reduced duration compared to MG1655)?*

To answer this question, we performed front migration, growth rate and single-cell tracking measurements on RP437. We found that this strain had nearly identical run-tumble statistics to our founder strain (MG1655-motile, Figure 1—figure supplement 3), a lower growth rate and a nearly two-fold lower rate of migration relative to founder. We conclude that our round 15 evolved strain is phenotypically distinct from the wild-type for chemotaxis laboratory strain RP437.

*Reviewer #3:*

*1) In general, I enjoyed the writing style and the attention to detail in the manuscript. In particular, the experiments were solid and the controls were thorough, especially for Figure 2. However, I have trouble understanding why the authors did not propagate all lines to 15 rounds since it only involves doing 5 more plates for each line. This is especially important since a lot of subsequent analysis is carried out using the "evolved strain" that harvested from that single line of evolution after 15 rounds. Having more lines would also indicate whether or not the reduction in diversity is a general feature.*

We have addressed this concern in two ways. (1) we have performed an additional set of four fully independent selection experiments out to 15 rounds. (2) We have performed single-cell tracking on intermediate rounds of selection (5 and 10) as well as on three independently evolved round 15 strains. For selection in the minimal medium environment we have performed five replicate selection experiments as well, sequencing intermediate strains and performing tracking on independently evolved round 10 strains.

*2) The modeling provides nice qualitative support to the findings about the relationship between front speed and geometry and the cells parameters but maybe takes too much place in the paper since many of the parameters were poorly constrained or taken from experiments done on other strains in different conditions. The authors are up front about this, and do a good job explaining the model, but it seems that the model is just used to say that the observed changes are probably not due to changes in growth (which they show experimentally anyway), and that there are probably other factors not accounted for in the model that are responsible for improved performance. Space could be used instead for addressing concerns mentioned above.*

Since the model now plays an important role in pointing out that the phenotypic trade-off we observe constrains the evolution of faster migration, we feel it is important to include a somewhat detailed discussion. We have made every effort to make the model accessible, reduce the length of the description of the model and have included a table in the main text describing each parameter.

*3) In Figure 1 the profile of intensity appears to get gradually lower from the center, and then abruptly lower to form the outer ring. I don't see how this can be compared to the simulated profile, which appears to gradually increase in intensity from the center, before forming an abruptly higher intensity ring. Is this why the profiles in Figure 3—figure supplement 4 look so different from one another? Can you account for these differences? Is it due to uneven illumination in the experiment? Did the authors use an inverse scaling for the Figure 4 (dark is more cells rather than light is more cells)?*

The outer ring is formed by migration towards serine and the secondary ring by migration towards aspartic acid. In the founder strain, the second ring is higher density and close to this outer ring and this results in the profile mentioned by the reviewer. The model includes only a *single* nutrient source (serine) and neglects the secondary ring formed by chemotaxis towards aspartic acid entirely. This accounts for the difference between the experimental and simulated front profiles. We have added Figure 2—figure supplement 1 to show the front profile for the founder strain more clearly. In all figures showing colonies, darker areas correspond to more cells.

*4) In the Results, the authors say that "the increase in s was reproducible across independent selection experiments", but it is hard to tell without showing the trajectory of each experiment. Figure 2 shows that on average, s increased. This is especially interesting because in Figure 3, the round where one experiment is missing has a much smaller error bar than the other rounds. This suggests that one of the experiments was quite different from the others. Was the missing observation from the experimental replicate that had a different mutational trajectory?*

To clarify the presentation of the data we have plotted each selection experiment as a separate line in Figure 1. We have included the four new 15-round selection experiments. This presentation substantially clarifies the interpretation of the data. For clarity, we have omitted the previously performed experiments that were only carried out to 10 rounds in rich medium. For these strains we performed sequencing and found very similar mutational trajectories to what we observed in the new 15-round experiments.

*5) Since tracks were cut when cells interacted with the boundary, how where the runs that were not terminated on both ends by tumbles treated (or tumbles that were not terminated on both ends by runs)? These runs (tumbles) would be of undefined length. Was the data analyzed without these undefined events?*

The data was analyzed without these undefined events. Run events that resulted in interaction with the chamber boundary were not included in the distributions shown in Figure 3 or any of the supplementary figures.

*6) Variance in the S dataset is in general greater than it is for the L dataset. The authors explain that this is due to sampling fewer run-tumble events for each individual in the S dataset. From the description, S individuals were observed for ~5 min, while L individuals were observed for ~10 min. However, there were at least 6x more individuals in S. Could the larger population size account for the increased variance? In general to assess cell-to-cell variability one should measure many cells rather than a few cells for a very long time. On a related note, the reduction in variance seems to be only significant (according to Figure 6 for one of the two datasets in two of the three measured parameters for which a reduction in variance is claimed (S for run speed, L for run time). Ideally, the significance of the reduction should not depend on the dataset.*

First, to better test for changes in individuality we performed single-cell tracking of ~100 individuals from strains isolated after 5,10 and (3 independent isolates) 15 rounds. From these data we observe unambiguous reduction in cell to cell variability (For example, see Figure 7). As stated above, since the change in individuality is no longer a focus of the present study we have omitted these results from the manuscript under consideration.

[Editors' note: the author responses to the re-review follow.]

*[…] Reviewer #1:*

*Although tradeoff between run speed and growth rate has been observed before, I do like the contrast between different outcomes in different environments. Overall, I find the paper solid and interesting. However, their narratives can be modified to achieve a greater level of clarity. I list a few examples below.*

*1) Authors claimed that in both rich and minimal media, their five lines evolved similarly. I am not sure that I agree with their assessment, although without error bars in graphs (Figure 1), it is tough to tell one way or another. I suggest authors add error bars (or an estimation of errors). Authors could also simply write something like "migration rates increased in all lines, though the extent of improvement differed along lines".*

We do not include error bars in Figure 1 since the error of the regression used to determine front migration rate are smaller than the size of the markers. We have clarified this in the caption to Figure 1.

To further emphasize that the increase in migration rate varies between replicates we have added the statement “So, while migration rates increased in all replicates, the magnitude of the increase differed between replicates.” to the end of paragraph 3 in the section “Experimental evolution of migration rate”.To address this issue in the minimal medium experiment we have added the statement “So while all replicates increased their migration rate, the magnitude of the increase in migration rate varied substantially.” to paragraph 6 of the aforementioned section.

*2) In some lines, despite selection, migration rate seemed to decline at later stages. Is that caused by genetic drift, as described by for example "Genetic drift at expanding frontiers promotes gene segregation" by Hallatschek et al.? This is a striking feature of the graph, and in my opinion, should be touched upon even if briefly.*

A substantial fraction of the observed fluctuations in migration rates is due to uncontrolled variation in the agar concentration of the plates. This variation arises from variation in liquid loss during autoclaving and from evaporation after plates are prepared. We have taken every measure to minimize these effects (autoclaving in large bottles, para-filming plates, storing plates at 4°C prior to use). We have added these details to the Methods: “Motility Selection”.By weighing plates we estimated the magnitude of the change in agar concentration due to evaporative loss. These experiments and previous measurements of the migration rate as a function of agar concentration (Croze *et al.* 2011, (Figure 4)) we expect evaporative losses can account for a change in migration rate in rich media of at most 0.06 cm/h. We have added these estimates to Supplementary file 1.

Therefore, only differences greater than approximately 0.06 cm/h in the rich medium experiment can be interpreted as being significant. For example, replicates 3 and 5 (Figure 1) have significantly faster rates of migration than other replicates in round 7. Further, the decline in migration rate in replicate 4 at round 14 and 15 is significant. These differences may reflect the distinct mutations arising in replicate experiments as observed in Figure 5. To address this we have added the following language to the end of paragraph 3 of “Experimental evolution of migration rate”:

“We estimate that plate-to-plate variation in agar concentration due to evaporative loss could change the migration rate by up to 0.06 cm/h in later rounds (Supplementary file 1). […] So, while migration rates increased in all replicates, the magnitude of the increase differed between replicates.”

For minimal medium conditions we have included similar estimates of the role of evaporation in changing migration rate to Supplementary file 1. We find that evaporative losses can account for about 10% variation in migration rates. Since the replicate to replicate migration exceeds this estimate we have added the following statement to paragraph 6 of “Experimental evolution of migration rate”:

“We observed an approximately 3-fold increase in s over the course of 10 rounds of selection. […] This variation may be due to the different mutations present across replicates (Figure 5).”

For clarity we have moved the sentence “Populations formed small ~1.5cm diameter colonies […]” that was previously at the end of this paragraph to the third sentence.

Four lines of evidence suggest that the variation in migration rates we observe cannot be accounted for by neutral variation as described in Hallatschek et al. (PNAS, 2007). First, Hallatschek et al.describe spatially segregated *neutral* diversity in expanding colonies on hard agar plates. This diversity does not confer a fitness advantage or disadvantage in the colony expansion process (as observed through competition experiments and colony morphology). Second, we sample from 8 locations around the periphery of the colony after each round of expansion and mix these samples prior to inoculating the next round. Therefore, spatially segregated diversity is destroyed after each round of selection. We have emphasized this by adding “After a fixed period of incubation samples are taken from 8 locations around the outer edge of the expanded colony, mixed, and used to inoculate a fresh plate.” to the caption for Figure 1. Third, we transfer a relatively large population (10^6^ cells) from one round to the next which limits genetic drift. Finally, while there is variation in the identity of mutations we observe (Figure 5), repeated mutations at identical nucleotides (*clpX*E185*) or genes (*galS*) suggest that selection is stronger than drift in our experiment.

*3) You often claimed that your model qualitatively captured experiments (e.g. Figure 2—figure supplement 1). I am not sure which qualities you are referring to, especially with respect to rich medium experiments. The agreement is pretty poor in my opinion, unless I am missing something.*

To more clearly describe the limitations of our model, and to clarify why the model predictions do not entirely recapitulate the colony expansion in rich medium in particular, we have added paragraph 4 to the section “Increasing swimming speed and growth rate increase migration rate”, which reads:

In rich medium our model describes the dynamics of a single metabolite/attractant (L-serine), and therefore fails to account for secondary fronts behind the outermost front, which arise from the metabolism of other amino acids (Adler, 1966)(Figure 1, Figure 1—figure supplement 2). […] While more sophisticated models have been developed to include these processes (Vladimirov et al.2008,Frankel et al. 2014), the model in ([Disp-formula equ1 equ2]) captures the essential features of bacterial front migration with fewer adjustable parameters. See the Supplementary file 1 for further discussion.

We have added a brief statement to the end of paragraph 2 in section “Experimental evolution of migration rate**”** to clarify that in minimal medium we observe only a single migrating front “Second, we used minimal medium (M63, 0.18mM galactose, 0.3% w/v agar, 30C) where populations migrate towards and metabolize galactose *with a single migrating front.*”

We have clarified the interpretation of the parameters in the reaction-diffusion model by adding “k_0_ is the chemotactic coefficient, which captures the strength of chemotaxis in response to gradients in attractant.” To the first paragraph of the “Increasing swimming speed and growth rate increase migration rate” section.

*Reviewer #2:*

*[…] Evolutionary dynamics under multiple selection pressures have been investigated by a number of studies, some of which were cited in the manuscript. The main novelty of this paper, as the authors pointed out, is in that the selection process here involves multiple stresses simultaneously. In this regard, I hope the following point could be clarified: By selecting populations at migration front, one clearly imposes selection pressure on migration rate; but it is less clear to me whether a selection pressure on growth rate exists. Having a higher or lower growth rate does not necessarily guarantee whether a subpopulation can reach the edge, or for the specific population at colony edge, growth rate and migration rate could be uncoupled. Escherichia coli colony spreading through soft agar indeed depends on both motility and growth, but this is at the level of entire colony.*

We argue that selection for faster migration also selects for faster growth. Growth results from nutrient consumption which creates a spatial gradient in nutrients (attractants) which then drives chemotaxis, migration and further growth. The coupling between growth and migration is precisely the reason that our model predicts a substantial increase in migration rate with growth rate in both conditions (Figure 2). So our numerical simulations support this claim.

To further support this claim we point the reviewer to work by D.A. Koster et al.(J. Mol. Biol. 2012, 424 180-191). The authors study migrating fronts of *E. coli* in low viscosity agar and show that cells in the migrating front can be modeled as a population in a chemostat. Fresh nutrients are provided by front migration while cells which lag behind the migrating front reside in low nutrient conditions and are “washed out” of the front. Cells that fail to grow at the front, or that grow at a reduced rate, will be outcompeted exponentially fast. Therefore, as our own model concludes, selection for migration rate also selects for growth rate.

Finally, our data also suggest a relationship between growth rate and migration rate. For the RP437 strain we measure nearly identical swimming properties as compared to MG1655-motile (run speed and duration; Figure 1—figure supplement 3). However, RP437 exhibits a growth rate which is approximately 10% lower than MG1655-motile and a migration rate in rich medium a factor of two lower than MG1655-motile. The slower migration rate, we postulate, results from the slower growth rate in this strain. We have added the migration rate and growth rate in rich medium for RP437 to the caption to Figure 1—figure supplement 3.

To further clarify the dual roles of growth and motility to the migration process we have added the following sentence to the first paragraph of the section “Experimental evolution of migration rate”. *“*As a result, the outermost edge of the expanding colony is driven by both growth and motility (Koster et al. 2012).”

*Another way to impose selection pressures simultaneously on growth rate and migration rate would be: (a) grow many plates in parallel; (b) select the plate with largest colony size; (c) pool all cells on that plate for next round of propagation and selection. (Reviewing editor's comments: you can discuss that in Discussions).*

We agree with the reviewer that the precise method of selection is important for interpreting results. We also agree that adding a discussion of this makes an important improvement to our manuscript. As we outline above, our method of selecting cells from the migrating front selects for both motility and growth rate. However, previous studies have invoked selection methods similar to those outlined by reviewer #2 (e.g. van Ditmarsch et al. 2013). To address the role of the selection procedure in the phenotypic outcome, we have modified paragraph 4 of the Discussion section as follows:

“Previous experimental evolution studies have revealed similar trade-offs to those presented here. […] A more precise understanding of the selection pressure applied by van Ditmarsch et al. might emerge from the application of Lande's (Lande, 1979) formalism to the observed genetic and phenotypic variation.”